



# Magnesium (Mg/Ca, $\delta^{26}$Mg), boron (B/Ca, $\delta^{11}$B), and calcium ([Ca$^{2+}$]) geochemistry of *Arctica islandica* and *Crassostrea virginica* extrapallial fluid and shell under ocean acidification

Blanca Alvarez Caraveo[1,2], Maxence Guillermic[1,2,3], Alan Downey-Wall[4], Louise P. Cameron[4], Jill N. Sutton[5], John A. Higgins[6], Justin B. Ries[4], Katie Lotterhos[4], Robert A. Eagle[1,2]

[1]Atmospheric and Oceanic Sciences Department, University of California, Los Angeles, Math Sciences Building, 520 Portola Plaza, Los Angeles, CA 90095, USA
[2]Center for Diverse Leadership in Science, Institute of the Environment and Sustainability, University of California, Los Angeles, LaKretz Hall, 619 Charles E Young Dr E no. 300, Los Angeles, CA 90024, USA
[3]Earth, Planetary and Space Sciences, Department, University of California, Los Angeles, Los Angeles, CA 90095, USA
[4] Department of Marine and Environmental Sciences, Marine Science Center, Northeastern University, 430 Nahant Rd, Nahant, MA 01908, USA
[5] Université de Brest, UMR 6539 CNRS/UBO/IRD/Ifremer, LEMAR, IUEM, 29280, Plouzané, France
[6] Department of Geosciences, Princeton University, Guyot Hall, Princeton NJ 08544, USA

*Correspondence to*: Blanca Alvarez Caraveo (alvarezblanca@g.ucla.edu) and Robert Eagle (robeagle@ucla.edu)

**Abstract.** The geochemistry of biogenic carbonates has long been used as proxies to record changing seawater parameters. However, the effect of ocean acidification on seawater chemistry and organism physiology could impact isotopic signatures and how elements are incorporated into the shell. In this study, we investigated the geochemistry of three reservoirs important for biomineralization - seawater, the extrapallial fluid (EPF), and the shell - in two bivalve species, *Crassostrea virginica* and *Arctica islandica*. Additionally, we examined the effects of three ocean acidification conditions (ambient: 500 ppm $CO_2$, moderate: 900 ppm $CO_2$, and high: 2800 ppm $CO_2$) on the geochemistry of the same three reservoirs for *C. virginica*. We present data on calcification rates, EPF pH, measured elemental ratios (Mg/Ca, B/Ca), and isotopic signatures ($\delta^{26}$Mg, $\delta^{11}$B). In both species, comparisons of seawater and EPF Mg/Ca and B/Ca, [Ca$^{2+}$], and $\delta^{26}$Mg indicate that the EPF has a distinct composition that differs from seawater. Shell $\delta^{11}$B did not faithfully record seawater pH and $\delta^{11}$B-calculated pH values were consistently higher than pH measurements of the EPF with microelectrodes, indicating that the shell $\delta^{11}$B may reflect a localized environment within the entire EPF reservoir. In *C. virginica*, EPF Mg/Ca and B/Ca, as well as absolute concentrations of Mg, B, and [Ca$^{2+}$], were all significantly affected by ocean acidification, indicating that OA affects the physiological pathways regulating or storing these ions, an observation that complicates their use as proxies. Reduction in EPF [Ca$^{2+}$] may represent an additional mechanism underlying reduction in calcification in *C. virginica* in response to seawater acidification. The complexity of dynamics of EPF chemistry suggest boron proxies in these two mollusc species are




not straightforwardly related to seawater pH, but ocean acidification does lead to both a decrease in microelectrode pH and
boron-isotope-based pH, potentially showing applicability of boron isotopes in recording physiological changes.
Collectively, our findings show that bivalves have high physiological control over the internal calcifying fluid, which
presents a challenge to using boron isotopes for reconstructing seawater pH.

# 1 Introduction

The elemental geochemistry of marine biogenic carbonate shells is widely used to track and reconstruct environmental
change (Broeker and Peng, 1982; Elderfield, 2006). The incorporation of elements within the skeleton of marine calcifiers
has been shown to be correlated with different environmental parameters, such as temperature (Dunbar et al., 1994, Alibert
and McCulloch 1997) and pH (e.g. Hemming and Hanson, 1992; Hönisch et al., 2004; McCulloch et al., 2018). However, it
has long been recognised that elemental and isotopic signatures of biogenic carbonate deviate from inorganic carbonate
grown under the same conditions, complicating the use and interpretation of these theoretical models for
paleo-reconstructions (e.g.. Urey, 1951; Craig, 1953; reviewed by Weiner and Dove, 2003). The physiological processes alter
the geochemistry of biominerals and consequently offset the environmental signal incorporated in biogenic carbonates,
termed "vital effects" (Urey, 1951) which includes the different biomineralization strategies that can modify the chemistry of
the calcification fluid (Weiner and Dove, 2003). For organisms to calcify, a semi-isolated calcification space will be, to
varying degrees, separated from seawater for supersaturation to be achieved in support of calcification (Weiner and Dove,
2003). In intracellular calcification, biominerals can be formed within cells using specialized vesicles or vacuoles, whereas in
extracellular cases, calcification may occur on an organic matrix template, with ions transported as necessary for crystal
nucleation to occur (Weiner and Dove, 2003; Addadi et al., 2006; reviewed by Gilbert et al., 2022). Additionally, the
geochemistry of the calcification fluid can be altered due to differing degrees of isolation from the parent fluid, seawater, as
well as the modulation of the calcification fluid chemistry via different methods of passive or active ion transport to the site
of calcification (Weiner and Dove 2003; McCulloch et al., 2017; Sutton et al., 2018; Liu et al., 2020). A mechanistic
understanding of such vital effects is desirable for the accurate interpretation of geochemical proxies preserved in the shells
of these organisms.
Molluscs have long been recognized as valuable archives for climate reconstructions, given the annual resolution growth
bands, long lifespans, and wide geographic distributions (Gibson et al., 2001; Peharda et al., 2021). However, it is also well
established that mollusc shell carbonates can express significant vital effects in many geochemical parameters (Schöne,
2008). For example, the $\delta^{11}B$ proxy for seawater pH in foraminifera and corals seems relatively insensitive in many molluscs
examined, including *Mytilus edulis, Mercenaria mercenaria*, and *Crassostrea virginica* (Heinemann et al., 2012; Foster and
Rae, 2016; McCulloch et al., 2017; Liu et al., 2020; Eagle et al., 2022). Shell B/Ca has been shown to be correlated to
internal fluid pH in *Mytilus edulis* (Heinemann, 2012) and *Mercenaria mercenaria* (Ulrich et al., 2021), but relationships to
seawater pH were less clear. Reported Mg/Ca are widely used as temperature proxies in many marine calcifiers



(Wannamaker 2008), however it is also long established that molluscs can regulate and actively exclude [$Mg^{2+}$] from their
shells (Lorens and Bender, 1977; Planchon et al., 2013), showing that biological regulation of biocalcification and the parent
fluids for shell formation can have a strong influence on Mg-based geochemical proxies. Mg isotope analyses can potentially
inform the [$Mg^{2+}$] transport process in molluscs. Although few Mg isotope studies of molluscs have been done, a study by
Planchon et al. (2013) investigated $\delta^{26}Mg$ across *Ruditapes philippinarum* tissues, shell, and fluid reservoirs and found that
seawater and extrapallial fluid magnesium signatures similar, suggesting that seawater is the source of [$Mg^{2+}$] ions within the
extrapallial fluid. Additionally, Planchon et al. (2013) found that Mg signatures within the shell varied between specimens
and were either in line with or deviated from inorganically precipitated aragonite, suggesting an ability for some clams to
physiologically alter or regulate [$Mg^{2+}$] within the extrapallial fluid.

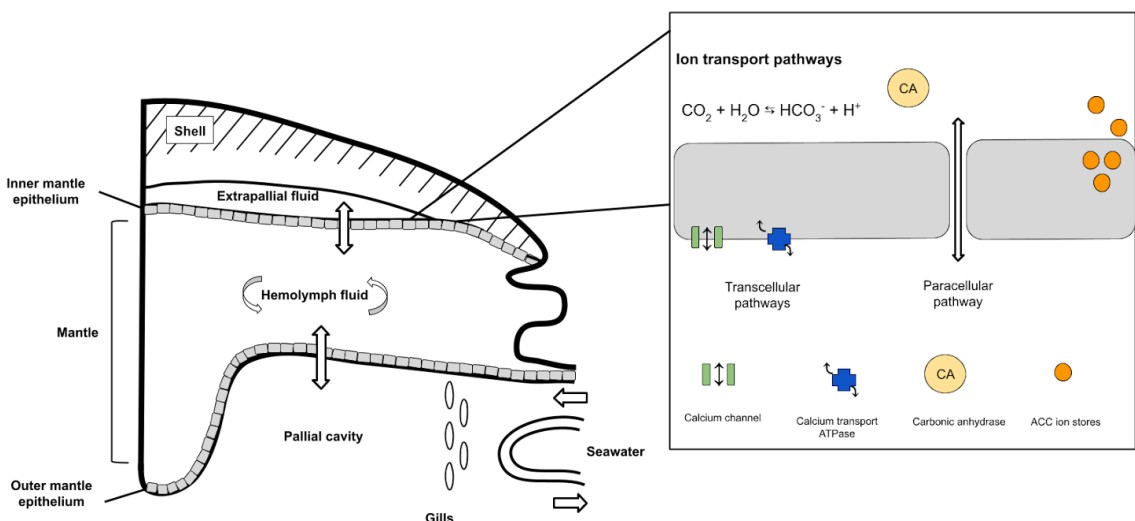

f 01



Figure 1. Schematic of a bivalve cross section showing the flow of between biomineralization ion reservoirs. The box on the
right shows a zoomed in schematic across the inner mantle epithelium cells that show transcellular and paracellular ion
transport pathways in and between epithelial cells. Figure adapted from Planchon et al. (2013) and Zhao et al. (2016).






Understanding the structure of mollusc tissues, internal fluid reservoirs, mechanisms of calcification and ion transport to the
site of calcification is critical to understanding these vital effects (Fig 1). It may also give insight into the sensitivity of
bivalves to $CO_2$-induced ocean acidification, a major environmental challenge to ocean ecosystems and commercial shellfish
fisheries (Gazeau et al., 2013; Stewart-Sinclair et al., 2020). Typically, bivalves are amongst the more sensitive group of
marine calcifier species to acidification (Ries et al., 2009; Kroecker et al., 2011).

| | Control *A. islandica* | Control *C. virginica* | Moderate OA *C. virginica* | High OA *C. virginica* |
|---|---|---|---|---|
| Measured seawater parameters | | | | |
| pH (total scale) | 7.93 ± 0.09 | 8.01 ± 0.08 | 7.75 ± 0.07 | 7.29 ± 0.11 |
| DIC (µmol/kg) | n/d | 1966 ± 44 | 1998 ± 212 | 2177 ± 160 |
| TA (µmol/kg) | n/d | 2120 ± 46 | 2120 ± 42 | 1511 ± 40 |
| Mg/Ca (mol/mol) | 5.13 ± 0.07 | 5.15 ± 0.07 | 5.23 ± 0.06 | 5.12 ± 0.03 |
| $\delta^{26}Mg$ (‰) | -0.82  0.06 ‰ | -0.77 ± 0.01 | -0.82 ±0.03 | -0.76 ± 0.09 |
| B/Ca (mol/mol) | 41.75 ± 1.52 | 41.66 ± 1.07 | 43.08 ± 2.9 | 42.11 ± 1.8 |
| $\delta^{11}B$ (‰) | 39.88 ± 0.13 | 40.29 ± 0.33 | 39.39 ± 0.33 | 39.82 ± 0.33 |
| Calculated seawater parameters | | | | |
| $p$CO$_2$ (ppm) | n/d | 570 ± 90 | 990 ± 173 | 2912 ± 373 |
| $[CO_3^{2-}]$ (µM) | n/d | 120 ± 12 | 79 ± 13 | 31 ± 4 |
| $\Omega_{Calcite}$ | n/d | 2.95 ± 0.30 | 1.93 ± 0.32 | 0.75 ± 0.09 |
| $\Omega_{Aragonite}$ | n/d | 1.89 ± 0.19 | 1.24 ± 0.21 | 0.48 ± 0.06 |
| $\delta^{11}B$-calculated EPF pH (total scale) | 7.76 ± 0.07 | 8.12 ± 0.09 | 8.06 ± 0.10 | 8.01 ± 0.08 |





| | | | | |
|---|---|---|---|---|
| $\triangle pH_{SW\text{-}\delta 11B,pH}$ | 0.17 | 0.64 | 0.77 | 0.8 |
| EPF geochemistry | | | | |
| microelectrode EPF pH (total scale) | 7.41 ± 0.14 | 7.48 ± 0.15 | 7.29 ± 0.10 | 7.21 ± 0.10 |
| $\triangle pH_{SW\text{-}EPF}$ | 0.52 | 0.53 | 0.46 | 0.08 |
| Mg/Ca (mol/mol) | 4.25 ± 0.67 | 4.55 ± 0.50 | 5.73 ± 0.34 | 5.58 ± 0.46 |
| $\delta^{26}Mg$ (‰) | -0.69 0.01 ‰ | -0.88 ± 0.06 | -0.87 ± 0.07 | -0.9 ± 0.1 |
| B/Ca (mol/mol) | 31.17 ± 4.87 | 33.66 ± 2.81 | 42.22 ± 3.33 | 43.26 ± 2.82 |
| $\delta^{11}B_{EPF}$ (‰) | 39.5 ± 0.4 | 39.3 ± 1.0 | 38.9 ± 0.47 | n/d |
| Shell geochemistry | | | | |
| Mg/Ca (mmol/mol) | 0.8 ± 0.2 | 13.8 ± 1.7 | 13.4 ± 2.3 | 12.3 ± 1.5 |
| $\delta^{26}Mg$ (‰) | n/d | -3.2 ± 0.1 | -3.1 ± 0.1 | -3.0 ± 0.2 |
| B/Ca (μmol/mol) | 57 ± 17 | 114 ± 22 | 125 ± 11 | 124 ± 9 |
| $\delta^{11}B_{Shell}$ (‰) | 15.26 ± 0.41 | 18.34 ± 0.59 | 16.91 ± 0.56 | 16.84 ± 0.35 |


Table 1. Seawater and extrapallial fluid carbonate chemistry parameters (pH, DIC, TA, Ω, δ11B-calculated EPF pH, and △pH) for both C. virginica and A. islandica under control conditions and C. virginica for OA conditions.. Seawater, extrapallial fluid, and shell geochemical parameters (Mg/Ca, δ26Mg, B/Ca, δ11B) for both C. virginica and A. islandica under control conditions and C. virginica for OA conditions. Parameters that were not measured or calculated are marked with 'n/d.'


The bivalve mollusc extrapallial fluid (EPF) is an internal fluid reservoir physically semi-separated from seawater that circulates in the pallial cavity, between the outer mantle epithelium (OME) and shell. Seawater enters the pallial cavity when



valves are open, then the internal hemolymph fluid circulates within the organs of the mollusc and finally can also be
transported across the mantle to the EPF (Table 1; Zhao et al., 2018). Bivalve mollusc shell calcification is thought to occur
at the interface of the EPF and growing shell where the ions for calcification interact with organic matrices, such as
polypeptide molecules (Crenshaw, 1972; Wheeler and Sikes, 1984; Wilbur and Bernhardt, 1984; Addadi, 2006) and proteins
within the EPF that act as a scaffolding template for nucleation and are important in the calcification process (Crenshaw,
1972; Wilber and Bernhardt, 1984). Additionally, molluscs can calcify though a transient amorphous calcium carbonate
precursor phase in which disordered calcium carbonate crystals can be stored and then transported to the calcification front
(Addadi, 2003; Immenhauser et al., 2016), which can act as another source of potential geochemical vital effects. Therefore,
it is expected that EPF chemistry will differ from seawater and that knowledge of EPF geochemistry may inform our
knowledge of vital effects in bivalve molluscs.
Unlike the calcifying fluid reservoirs in most organisms, bivalve EPF has a large enough volume that it can be directly
sampled, allowing for direct measurements of the reservoir to compare with seawater geochemistry and elucidate in situ
changes in EPF chemistry. A foundational study by Crenshaw (1972) found that, in three mollusc species, the EPF
calcification fluid had a different chemical composition and pH from seawater and from the mollusc hemolymph fluid
(Crenshaw et al., 1972). Crenshaw, (1972) reported that EPF pH was significantly lower than seawater pH, that cationic
compositions of the EPF could also differ from seawater, and that the total C (including all species of dissolved inorganic
carbon) of the EPF was higher than that of seawater. Additionally, Crenshaw also showed that EPF calcium concentration
and pH co-varied significantly over time during the opening and closing of valves, or the ventilation cycle. When valves are
closed pH is lower and calcium concentration higher, resulting from dissolution of shell material and return of calcium to the
EPF (Crenshaw, 1972). A previous study on the king scallop, *Pecten maximus*, by Cameron et al. (2019) showed that EPF
pH was lower than seawater and also depended on $pCO_2$ and temperature. Ramesh et al., (2017) reported, using a
microelectrode approach, that pH and $[CO_3^{2-}]$ were elevated proximal to the growing shell in larval *Mytilus edulis* shells. In
the quahog *Arctica islandica*, Stemmer et al. (2019) reported synchronous short-term fluctuations in $[Ca^{2+}]$ and pH at the
outer mantle epithelium surface. They attributed this to active ion pumping across mantle epithelial cells, which created
significant differences between carbonate saturation and pH of the bulk EPF and the EPF close to the outer mantle
epithelium.
Boron proxies utilise boron speciation and isotope fractionation in seawater to reconstruct pH and $[CO_3^{2-}]$ of seawater from
the chemistry of calcium carbonate shells (Hemming and Hanson, 1992; Hönisch et al., 2004). In seawater, the speciation of
boric acid $[B(OH)_3]$ and borate ion $[B(OH)_4^-]$ varies as a function of pH (Hemming and Hanson 1992). In addition to the pH
dependence of their relative abundances, the boron proxy also makes use of a large isotopic fractionation between the two
boron species (Klochko et al., 2006, Nir et al., 2015). A key assumption of the proxy is that boron, in the form of borate ion,
is the predominant form incorporated into the crystal lattice of calcite via carbonate ion substitution during the precipitation
of calcium carbonate (Hemming and Hanson 1992). The $\delta^{11}B$ of the carbonate ($\delta^{11}B_{CaCO3}$) should then, in theory, reflect the
boron isotopic composition of the borate ion in seawater ($\delta^{11}B_{CaCO3}$). Accurate reconstruction of seawater pH can then be



achieved using specific empirical relationships between the $\delta^{11}B_{CaCO3}$ and $\delta^{11}B_{CaCO3}$, which can in turn be used to determine
pH. The marine boron system is also utilized in the development of B/Ca proxies, which utilize the substitution of boron for
$[CO_3^{2-}]$ in the crystal lattice and the relationship between the partition coefficient ($K_D$), B/Ca, and $[CO_3^{2-}]$ to create a proxy
for $[CO_3^{2-}]$ of seawater or calcifying fluid (reviewed by DeCarlo et al., 2018). Using the exchange reactions for the
substitution of boron during aragonite or calcite precipitation, the founding assumption of the proxy is that B/Ca of the shell
can be used to calculate the $[CO_3^{2-}]$ of the solution from which the aragonite or calcite precipitated. Inorganic aragonite
precipitation experiments have validated the B/Ca proxy by allowing for the calculation of the partition coefficient ($K_D$)
between aragonite and seawater and fitting of experimental B/Ca data (Mavromatis et al., 2015; Holcomb et al., 2016;
Allison 2017; reviewed by DeCarlo et al., 2018). However the B/Ca proxy also has limitations, as it has only been developed
for aragonite samples and because of remaining unresolved differences in the formulation of the $K_D$, exchange reactions, and
fitting of B/Ca experimental data between studies (Allison et al., 2017; McCulloch et al., 2017; DeCarlo et al., 2018;
Holcomb et al., 2016). Together, both $\delta^{11}B$ ($pH_{CF}$) and B/Ca ($[CO_3^{2-}]$) proxies can be used to constrain the full carbonate
system of the calcifying medium (DeCarlo et al., 2018).
Vital effects of the $\delta^{11}B$ can be species-specific. In the case of foraminifera, vital effects are relatively minor (Hönisch et al.,
2004; Foster and Rae, 2016). However, other calcifying organisms, such as corals, coralline red algae, and molluscs, show
significant $\delta^{11}B$ deviations from relationships predicted from theoretical calculations (e.g.. Donald et al., 2017; Schoepf et al.,
2017; McCulloch et al., 2018; Sutton et al. 2018, Anagnostou et al., 2019; Liu et al., 2020). There are different theories to
explain the divergence of $\delta^{11}B$ from the seawater theoretical model. It is hypothesized for some taxa that $\delta^{11}B$ may not
faithfully record seawater pH, but rather the pH of the discrete fluid from which ions are sourced for calcification that may
be isolated or semi-isolated from seawater (Gilbert et al., 2022). Previous work on corals has used the boron proxy analyses,
along with other approaches, to probe internal carbonate chemistry of the calcification fluid (Ries, 2011; Holcomb et al.,
2014; Guillermic et al., 2021; Cameron et al., 2022; Eagle et al., 2022; Allison et al., 2023). All approaches, both
geochemical and physiological, indicate that corals elevate the pH and $[CO_3^{2-}]$ of their calcifying fluid to induce
calcification, but this mechanism is sensitive to ocean acidification and has yet to be fully understood (Liu et al., 2020;
Guillermic et al., 2021; Cameron et al., 2022; Eagle et al., 2022).
Beyond corals, few taxa have been studied using combined geochemical tracer work to determine the chemistry of
calcification fluid pools and sources of ions to the calcification front. Work by Sutton et al. (2018) noted that $\delta^{11}B$ values in
urchin spines were lower than seawater borate $\delta^{11}B$. Stumpp et al. (2013) showed that the internal pH of sea urchin larvae
was typically lower than seawater pH. Short et al. (2015), Donald et al. (2017), Anagnostou et al. (2019), and Liu et al
(2020) found high $\delta^{11}B$ in calcite produced by coralline algae, which is potentially consistent with elevation of calcifying
fluid pH in support of calcification either through enzymatic proton removal and/or photosynthetically driven removal of
dissolved inorganic carbon from the calcifying fluid. To date, one study has investigated the B/Ca and $\delta^{11}B$ of shell and EPF
of the bivalve *Mytilus edulis* (Heinemann et al., 2012).



The mollusc extrapallial fluid is an attractive target to investigate geochemical vital effects because not only can it be probed
with electrodes, like for corals, but it can also be extracted and analyzed. In this study, we investigate the $\delta^{11}$B, B/Ca, $\delta^{26}$Mg,
and Mg/Ca in extracted extrapallial fluid and aragonite shell of the quahog, *Arctica islandica*, and the calcite shell of the
eastern oyster, *Crassostrea virginica*. This allows for the investigation of the tripartite fractionation between seawater,
extrapallial fluid, and shell. Individuals were grown in controlled laboratory experiments, with extrapallial fluid pH
determined with microelectrodes, and other physiological parameters, such as calcification rate and tissue production,
determined by conventional methods (Downey-Wall et al., 2020). Specimens of *C. virginica* were also cultured in three
different treatments of $pCO_2$: ambient, moderate and high ocean acidification conditions. Geochemical analysis of the
seawater, shell, and extrapallial fluid thereby allow novel insights into the transport of ions from seawater to the extrapallial
fluid, and the fractionation of isotopes and elements between the extrapallial fluid and shell under both control and acidified
conditions.

## 174 2 Materials and Methods

### 175 2.1 Experimental Conditions

A detailed explanation of the collection and culturing of *C. virginica* and *A. islandica* is outlined in Downey-Wall et
al. (2020). Seawater salinity, temperature, and pH (total scale) were monitored and maintained throughout the experiment.
Seawater was maintained at a pH of 8.01 ± 0.08, temperature of 18.2 ± 1 °C, and salinity of 31 psu for the calcitic oyster *C.*
*virginica*. Seawater was maintained at a pH of 7.93 ± 0.09, temperature of 18.2 ± 1 °C, and salinity of 35 psu for the
aragonitic clam *A. islandica* in the control conditions (Downey-Wall et al., 2020).

Adult *C. virginica* specimens were collected from three intertidal sites on Plum Island Sound, Massachusetts, USA
(Site 1, 42.75 N, -70.84 E; Site 2,: 42.73 N, -70.86 E; Site 3, 42.68, -70.81) and transferred to Northeastern University's
Marine Science Center. Following a 33-day period of acclimation to laboratory conditions, oysters from each collection site
were exposed to control (mean $pCO_2$ ± SE = 570 ± 14 ppm; $\Omega_{calcite}$ = 2.95 ± 0.30 ), moderate OA (990 ± 29 ppm, $\Omega_{calcite}$ = 1.93
± 0.32), or high OA (2912 ± 59 ppm, $\Omega_{calcite}$ = 0.75 ± 0.09) treatments. Target $pCO_2$ treatment was achieved by mixing
compressed $CO_2$ and compressed ambient air using solenoid-valve-controlled mass flow controllers at flow rates that target
$pCO_2$ conditions. The treated seawater was introduced to the flow-through aquaria at a rate of 150 mL min$^{-1}$. Tank salinity,
temperature, and DIC and TA were measured for the duration of the experiment and used to calculate pH (total scale),
$\Omega_{calcite}$, $[CO_3^{2-}]$, $[HCO_3^-]$, $[CO_2]$, and $pCO_2$ of each tank using CO2SYS version 2.1 (Pierrot et al. 2011; see Downey-Wall et
al. 2020). Measured and calculated seawater parameters are reported in Table 1. Oysters were fed 1% Shellfish Diet 1800®
twice daily following best practices outlined in Helm and Bourne (2004).



## 2.2 Calcification rate measurements

Net calcification rate was calculated using the dry weight at the start and end of the experiment. Initial dry weight was measured at the start of exposure, on day 33 or 34, after the acclimation period (Downy-Wall et al., 2020). The buoyant weight was measured on either day 50 or 80 and the final dry weight was derived using a linear relationship between oyster dry weight and oyster buoyant weight (Ries et al., 2009).

## 2.3 Extrapallial fluid sampling

Sampling of the extrapallial fluid (EPF) was previously described in Downey-Wall et al. (2020). Briefly, a hole was drilled onto the shell to expose the EPF cavity, a port was inserted and sealed with epoxy to directly sample the EPF with a syringe and prevent seawater intrusion. Oysters recovered for 4 days before being transferred to experimental tanks for acclimation before the experiment. To sample the EPF, oysters were removed from the tanks and EPF was extracted by inserting a sterile 5 mL syringe with a flexible 18-gauge polypropylene tip through the port. EPF samples were stored in 2 mL microcentrifuge tubes and refrigerated at 6˚C for further analysis. pH (Total scale) of the EPF was measured directly after extraction using a micro-pH probe. EPF measurements were collected at the end of the experiment, on day 71, for *C. virginica* and day 14 for *A. islandica*. EPF pH diel variability was also explored by measuring EPF pH at 6 timepoints to produce time series for both species in a 24-hour period.

## 2.4 Shell sampling

Following EPF extraction, oysters were shucked and cleaned in 90% ethanol. The cleaned shells were dried at room temperature for 48 hours and sealed in plastic bags for analysis. For skeletal geochemical and elemental ratio analysis, the inner (lamellar) layer of the oyster shell was gently shaved with a diamond-tipped Dremel tool and about 5 mg of ground powder was stored in sealed microcentrifuge tubes.

## 2.4 Elemental ratio analysis

For the shells, about 2.5 mg of powder was sub-sampled from each specimen shell and cleaned with a 0.3 % hydrogen peroxide in 0.1 N sodium hydroxide solution to remove organic matter as described in Barker et al. (2003). Carbonate samples were dissolved in 1 N double-distilled HCl (see Guillermic et al., 2021, for details). Elemental ratios were measured on a Thermo Fisher Scientific Element XR HR-ICP-MS at the PSO (Plouzané, France) after Ca analyses on an Agilent ICP-AES Varian 710 at the University of California, Los Angeles (UCLA, Los Angeles, USA). Data quality and external reproducibility were maintained and quantified via repeated measurements of international standard JC$_P$-1 during a



particular session (Gutjahr et al., 2021). Typical measured concentrations of procedural blanks for the trace element analyses for sessions in which samples are diluted to 30 ppm Ca are $^7Li < 3\%$, $^{11}B < 4\%$, $^{25}Mg < 0.1\%$, $^{87}Sr < 0.1\%$, and $^{43}Ca < 0.1\%$. Typical analytical uncertainties on the X/Ca elemental ratios are 0.3 μmol/mol for Li/Ca, 21 μmol/mol for B/Ca, 0.09 mmol/mol for Mg/Ca, and 0.01 mmol/mol for Sr/Ca (2 SD, n = 28).

For EPF and seawater samples, 10 μL of sample was added to 490 μL of a solution of 0.1 N $HNO_3$/0.3 M HF. Mono-elemental solution of indium was added to reach a concentration of 1 ppb to monitor any matrix effect or drift of the instrument during a particular session. Standards were prepared by diluting an in-house seawater standard spiked with indium. International standards NRC-NASS-6 was used to ensure quality of the data.

## 2.5 Boron isotope analyses

Boron purification for the different samples was achieved via microdistillation following the method described in Guillermic et al. (2021) and originally developed by Gaillardet et al. (2001) and modified for Ca-rich matrix by Wang et al. (2010). 2.5-3.0 mg of oxidatively cleaned shell powders were dissolved in 1N HCl. For the EPF, 25 μL of EPF was added to 40 μL of 1N HCl. For the seawater, 50 μL of concentrated HCl was added to 450 μL of seawater. 60μL of each of the solutions was loaded for microdistillation. Boron isotopes were analyzed at the Pôle Spectrométrie Océan (PSO), Plouzané, on a Thermo Neptune inductively coupled plasma mass spectrometry (MC-ICP-MS) equipped with $10^{11}$ Ohm Faraday cup.

The certified boron isotope liquid standard ERM© AE120 ($\delta^{11}B$ = -20.2 ± 0.6 ‰, Vogl and Rosner, 2011) was used to monitor reproducibility and drift during each session. Samples measured for boron isotopes in carbonates were typically run at 80 ppb B (~30 ng B per <0.5 mL), whereas samples of EPF and seawater were typically run at 150-200 ppb B (~150 ng B per mL). Sensitivity on $^{11}B$ was 10 mV/ppb B (e.g., 10 mV for 1 ppb B) in wet plasma at 50 μL/min sample aspiration rate. Procedural boron blanks ranged from 0.3 to 0.4 ng B and the acid blank during analyses was measured at 3 mV on the $^{11}B$, indicating a total blank contribution of <2% of the sample signal with no memory effect within and across sessions. External reproducibility was ensured by the measurements of carbonate standard microdistilled at the same time as the samples. Results for the isotopic composition of the $JC_P$-1 is $\delta^{11}B$ =24.67 ± 0.28 ‰ (2 SE, n=41), within error of published values (24.36 ± 0.45 ‰, 2SD, Gutjahr et al., 2021).

## 2.6 Magnesium isotope analyses

Carbonate samples were dissolved in 0.1 N buffered acetic acid ammonium hydroxide solution over four hours in a sonicator. Samples were then centrifuged and aliquots of the supernatant were transferred into cleaned 15 mL centrifuge tubes. Aliquots of the bulk supernatants were then diluted ~30-fold and calcium and magnesium were separated and purified in different runs via a Thermo-Dionex ICS-5000+ ion chromatograph equipped with a fraction collector according to established methods outlined by Husson et al. (2015). EPF samples contained organics that obscured elution profiles, thus limiting the elemental yield and purification. Therefore, samples were digested on a hot plate in hydrogen peroxide and nitric



acid to remove organics prior purification. Seawater and EPF samples were purified through the Thermo-Dionex ICS-5000+
ion chromatograph using another elution method than for carbonate samples. Seawater and carbonate standards were also
purified at the same time to ensure quality of the method.

Samples were then dried and then rehydrated in a solution of 2% nitric acid. Magnesium isotopic ratios were
measured at Princeton University using a Thermo Neptune+ (MC-ICP-MS) spectrometer according to methods outlined in
Higgins et al. (2018) and Ahm et al. (2021). Samples were introduced via an ESI Apex-IR sample introduction system.
Magnesium isotope ratios ($^{26}Mg/^{24}Mg$) were measured in low resolution mode, with every sample bracketed by the analysis
of standards. Results are reported relative to the Dead Sea Magnesium-3 standard (DSM-3). Long term external precision on
magnesium isotope results at the Higgins Lab (Princeton) was determined through repeated measurements of the
Cambridge-1 standard (-2.59 $\pm$ 0.07‰, 2 SD, n = 19) and modern seawater (-0.82 $\pm$ 0.14 ‰, 2 SD, n = 21) and is reported
in Ahm et al. (2021). Measured standards during the analytical session are given for the Cambridge-1 standard (-2.60 $\pm$ 0.20
‰, 2 SD, n = 2) and for modern seawater (-0.82 $\pm$ 0.06 ‰, 2 SD, n=2).

## 264 2.7 Calculation of boron proxies and EPF carbonate chemistry

The use of boron proxies to reconstruct pH and [$CO_3^{2-}$] of the precipitating solution (i.e., the organism's calcifying
fluid) is based upon boron speciation and fractionation in seawater (Hemming and Hanson, 1992; Hönisch et al., 2004). In
seawater-type solutions, the speciation of boric acid [$B(OH)_3$] and borate ion [$B(OH)_4^-$] varies as a function of pH (Hemming
and Hanson 1992). In addition to the pH dependence of their relative abundances, the boron proxy also relies upon the large
isotopic fractionation between the two boron species (Klochko et al., 2006, Nir et al., 2015). A key assumption of the proxy
is that boron, in the form of borate ion, is the predominant form incorporated into the crystal lattice of calcite via carbonate
ion substitution during the precipitation of calcium carbonate (Hemming and Hanson 1992). The $\delta^{11}B$ of the carbonate
($\delta^{11}B_{CaCO3}$) should then, in theory, reflect the boron isotopic composition of the borate ion ($\delta^{11}B_{B(OH)4-}$) in the bivalve
calcifying fluid (extrapallial fluid), which in turn reflects pH of the calcifying (extrapallial) fluid.

The boron isotopic signature of the shell ($\delta^{11}B_{carb}$) was used to calculate pH of the calcifying fluid (pH$_{CF}$) using the
following equation (Hemming and Hanson, 1992; Zeebe and Wolf-Gladrow, 2001):

$$pH_{cf} = pK_B - \log\left(\frac{\delta^{11}B_{SW} - \delta^{11}B_{carb}}{\delta^{11}B_{SW} - \alpha * \delta^{11}B_{carb} - \varepsilon}\right)$$
           eq. 1


In equation 1, pK$_B$ is the dissociation constant, $\delta^{11}B_{sw}$ represents the measured boron isotopic composition of seawater,
$\delta^{11}B_{carb}$ represents the boron isotopic composition of the shell, and $\alpha/\varepsilon$ represents the boron isotopic fractionation factor/
fractionation between boric acid and borate ion (Klochko et al. 2006).



The saturation state of calcite ($\Omega_{cacite}$) and aragonite ($\Omega_{aragonite}$) of the EPF for each species were calculated using temperature,
salinity, pressure, measured EPF Ca, measured EPF Mg, pH either from microelectrode pH or $\delta^{11}$B-calculated pH, and
literature values of DIC (3000 for *A. islandica* from Stemmer et al. 2013, and 4200 for *C. virginica* from McNally et al.,
2022). The saturation states were calculated using Seacarbx with maximum input of [$Mg^{2+}$] allowed by the code for samples
presenting higher EPF [$Mg^{2+}$] than the limit allowed by the code (Raitzsch et al., 2021). Those saturation state values are
limited by the fact that no direct measurements of EPF DIC was performed during this study, and a range of [$Ca^{2+}$] and
[$Mg^{2+}$] values were measured in the EPF, resulting in a range of calculated saturation states as presented in Table 3.

**3 Results**
**3.1 Previous Culturing experiment, calcification rates, seawater chemistry, and EPF chemistry**



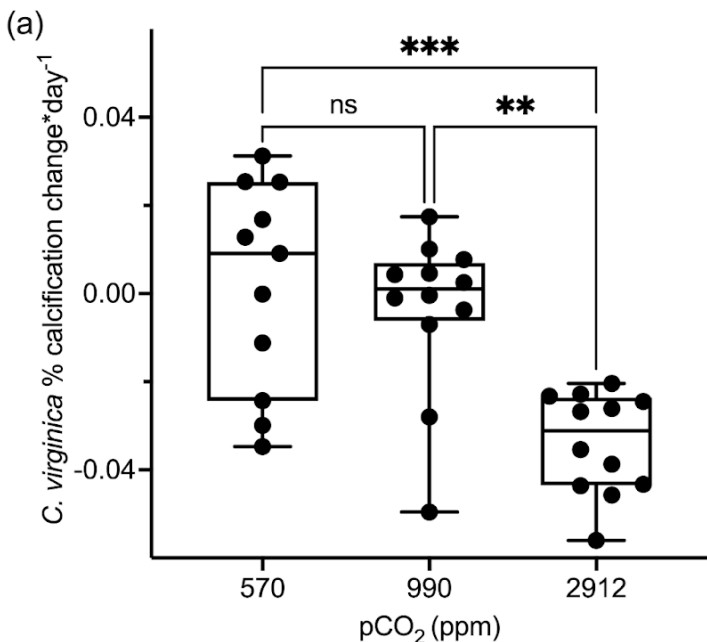

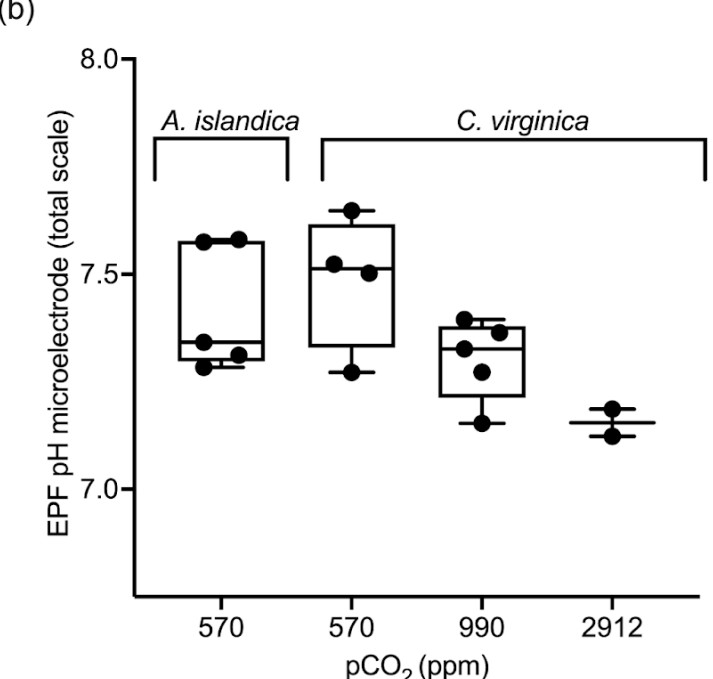

f02






Figure 2. (a) Box plots showing percent calcification change over the experiment for C. virginica for each treatment. Stars denote statistically different means and 'ns' signify non significant mean differences in a pairwise t-test (at significance p < 0.05). (b) Averaged microelectrode EPF pH for A. islandica under control conditions and C. virginica for OA conditions.

*Crassostrea virginica* specimens were previously cultured in experimental tanks with seawater that was continuously bubbled with gas mixtures comprising three $pCO_2$ levels (400 ppm, 900 ppm, 2800 ppm; see Downey-Wall *et al*., 2020). The highest $pCO_2$ treatment produced seawater values with a $\Omega_{calcite} < 1$, which does not favor calcification (Table 1). In this study, we present unpublished EPF pH microelectrode data for *A, islandica* cultured at a single control condition (400 ppm $pCO_2$) and we present published EPF microelectrode data for the *C. virginica* acidification experiment of Downey-Wall et al. (2020). Measured and calculated seawater parameters from the culture experiments are presented in Table 1. Percent change in calcification per day (Fig 2a), as well as EPF pH as measured by microelectrode (Fig 2B), decreased in *C. virginica* with increasing $pCO_2$ Both species had similar EPF pH (Fig 2b). Downey-Wall et al 2020 reported that *C. virginica* calcification decreased as $pCO_2$ increased and that, for each acidification treatment, the mean EPF pH during the experiment was lower than the corresponding seawater pH. Additionally, they report that using a linear model, $pCO_2$ treatment had a significant effect on EPF pH (linear model, p<0.05) and that at the highest $pCO_2$ treatment, EPF pH was significantly lower than seawater pH (Table 1; Fig 2; post hoc p-value<0.05 see Downey-Wall *et al*., 2020). We note that the *C. virginica* average ΔpH (seawater pH - EPF pH) decreased with decreasing seawater pH. The ΔpH for the control treatment was 0.53, the moderate OA treatment was 0.46, and the high OA treatment was 0.08. Here we report that at the control $pCO_2$ level, the EPF pH of *A. islandica* was 7.41, compared to 7.48 for *C. virginica* and the ΔpH for *A. islandica* was 0.52 (Table 1).





Figure 3. Box plots of Mg/Ca comparing seawater and extrapallial fluid for (a) C. virginica and (b) A. islandica, (c) comparing EPF Mg/Ca between species, and (d) shell Mg/Ca between species. Box plots of [Mg] comparing seawater and extrapallial fluid for (e) C. virginica and (f) A. islandica, (g) comparing EPF [Mg] between species. Box plots of [Ca] comparing seawater and extrapallial fluid for (h) C. virginica and (i) A. islandica, (j) comparing EPF [Ca] between species. Box plots of 26Mg comparing seawater and extrapallial fluid for (k) C. virginica and (l) A. islandica. Stars denote statistically different means and 'ns' signify non significant mean differences in a pairwise t-test (at significance $p < 0.05$). No comparison was tested on (l) due to limited sample size.

## 3.2 Mg/Ca of seawater, EPF, and bivalve shell

 

There was a significant decrease in EPF Mg/Ca compared to seawater Mg/Ca for both *A. islandica* and *C. virginica* (t-test,
n=2, p-value<0.05; Fig 3a-b). The Mg/Ca of *C. virginica* EPF was 4.55± 0.50 mol/mol and significantly higher than *A.*
*islandica* EPF which was 4.25±0.67 mol/mol (Fig 3d; Table 1). For both species, the low EPF Mg/Ca versus seawater
Mg/Ca was driven by higher $[Ca^{2+}]$ concentrations in the EPF relative to seawater (Fig 3h-i). Considering the elemental
concentrations alone, instead of as a ratio, there was no significant difference in EPF $[Mg^{2+}]$ or $[Ca^{2+}]$ concentrations between
species (Fig 3g and 3j). Shell Mg/Ca for the calcitic *C. virginica* was 13.8±1.7 mmol/mol and significantly higher than the
aragonitic *A. islandica* shell which was 0.8±0.02 mmol/mol, in line with shell polymorph mineralogy. The apparent partition
coefficient ($K_{Mg}$) between the seawater and the shell was 0.003 in *C. virginica* and 0.002 in *A. islandica* (Table 2). $K_{Mg}$
between EPF and shell was 0.003 in *C. virginica* and 0.002 in *A. islandica*. $K_{Mg}$ between seawater and the EPF is 0.9 for *C.*
*virginica* and 0.8 for *A. islandica* (Table 2).

| | | EPF/SW | | Shell/SW | | Shell/EPF | |
|---|---|---|---|---|---|---|---|
| | | *A. islandica* | *C. virginica* | *A. islandica* | *C. virginica* | *A. islandica* | *C. virginica* |
| $K_{Mg/Ca}$ | 400 | 0.8 | 0.9 | 0.0002 | 0.003 | 0.0002 | 0.003 |
| | 900 | | 1.1 | | 0.002 | | 0.002 |
| | 2000 | | 1.2 | | 0.002 | | 0.002 |
| $K_{B/Ca}$ | 400 | 0.7 | 0.8 | 0.001 | 0.003 | 0.002 | 0.003 |
| | 900 | | 0.9 | | 0.003 | | 0.003 |
| | 2000 | | 1.1 | | 0.003 | | 0.003 |


Table 2. Partition coefficients between EPF and seawater, seawater and the mineral, and EPF and the mineral for Mg/Ca and
B/Ca.
*C. virginica* seawater and EPF $\delta^{26}Mg$ were -0.77 ± 0.01 ‰ and  -0.88 ± 0.06 ‰, respectively and displayed a significant
decrease in EPF $\delta^{26}Mg$ compared to seawater for *C. virginica* (t-test, n1=3 n2=5, p-value< 0.05; Table 1, Fig 3k-l).  For *A.*
*islandica*, seawater and EPF $\delta^{26}Mg$ were -0.82 ± 0.06 ‰ and  -0.69 ± 0.01 ‰, respectively, but no statistical analysis could
be done between the two reservoirs owing to the small sample size (Table 1). The average shell $\delta^{26}Mg$ for *C. virginica* was



-3.2 $\pm$ 0.1‰, but *A. islandica* shell $\delta^{26}Mg$ could not be analyzed because of low shell $[Mg^{2+}]$ content and limited sample
material.





f 04



Figure 4. Box plots showing C. virginica (a) EPF Mg/Ca and (b) shell Mg/Ca across seawater pH treatments. Additionally, box plots show (c) EPF [Mg], (d) EPF [Ca], (e) EPF 26Mg, and (f) shell 26Mg. Stars denote statistically different means and 'ns' signify non significant mean differences in a pairwise t-test (at significance $p < 0.05$).

In the *C. virginica* acidification experiment, EPF but not shell Mg/Ca was found to increase as EPF pH decreased (regression, n=10, p-value<0.05; Fig 5a-b). OA treatment had a significant effect on shell Mg/Ca (ANOVA, n=10, p-value<0.05, Fig 4a-b). The concentration of both $[Ca^{2+}]$ and $[Mg^{2+}]$ in the EPF decreased with decreasing EPF pH (regression, n=10, p-value< 0.05; Fig 5c-d). However, when binning by seawater pH treatments, only the $[Ca^{2+}]$ and $[Mg^{2+}]$ of the ambient condition was significantly elevated compared to the moderate and high ocean acidification treatments (Tukey HSD, n1=4 n2=3, p<0.05, Fig 4c-d). The EPF and shell $\delta^{26}Mg$ did not change as a function of EPF or seawater pH (Fig 4e-f and 5e-f).





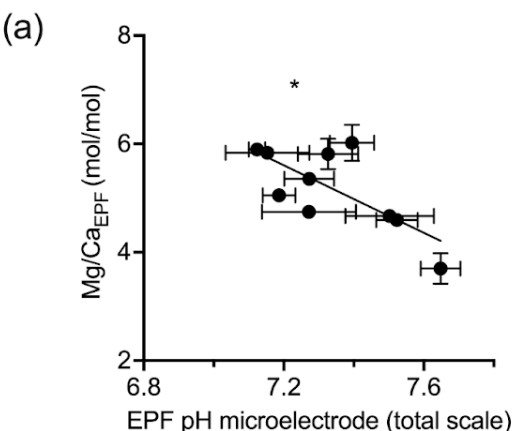

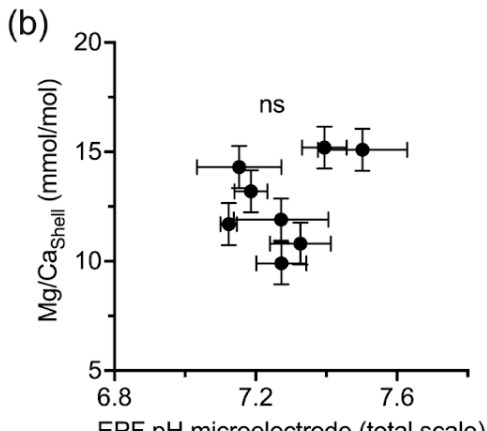

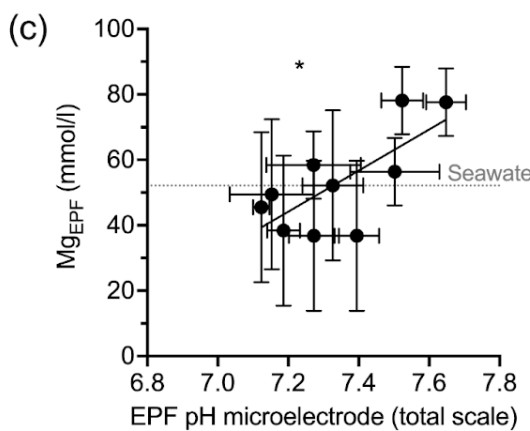

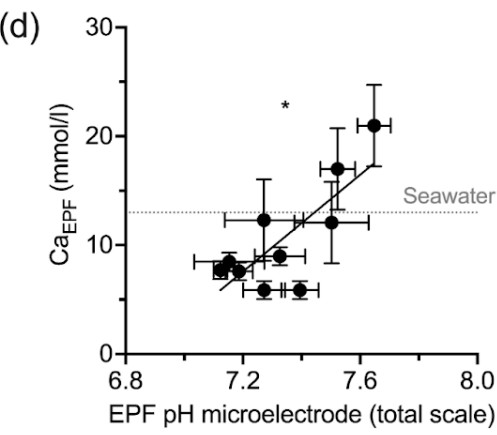

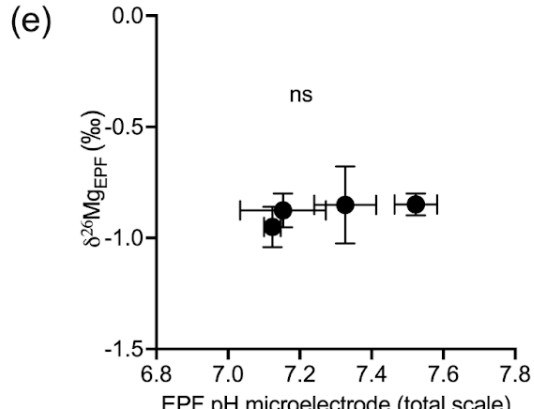

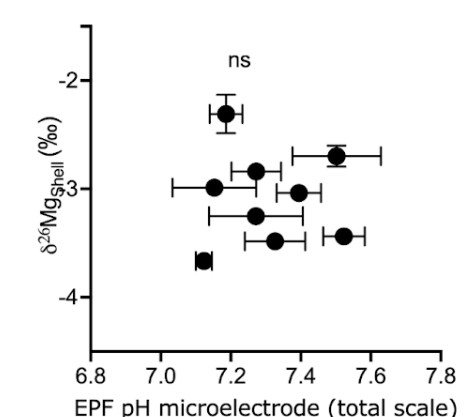

f 05




Figure 5. Scatter plots showing C. virginica individual specimen (a) EPF Mg/Ca and (b) shell Mg/Ca across corresponding
microelectrode pH. Additionally, scatter plots (c) EPF [Mg], (d) EPF [Ca], (e) EPF 26Mg, and (f) shell 26Mg across
microelectrode EPF pH. Stars denote statistically significantly nonzero regression slopes and 'ns' signify non significant
regressions (at significance $p < 0.05$). Dotted gray lines on (c) and (d) show the average [Mg] and [Ca] seawater
concentration, respectively.

**3.3 Boron geochemistry of seawater, EPF, and shell**

*A. islandica* EPF B/Ca was $27.91 \pm 4.87$ mmol/mol and was significantly lower than seawater B/Ca which was $41.75 \pm 1.52$
mmol/mol (t-test, n1=7 n2=5, p-value<0.05, Fig 6a). *C. virginica* EPF B/Ca was $41.66 \pm 1.07$ mmol/mol and was
significantly lower than seawater B/Ca which was $33.66 \pm 2.81$ mmol/mol (t-test, n1=6 n2=5, p-value<0.05 Fig 6b) The
boron concentration was not significantly different between seawater and EPF for both *C. virginica* and *A. islandica* (Fig
6e-f). There was no significant difference in shell or EPF B/Ca between *C. virginica* and *A. islandica* (Fig 6c-d). The
apparent partition coefficient ($K_B$) between the seawater and the shell was 0.003 in *C. virginica* and 0.001 in *A. islandica*. $K_B$
between EPF and shell was 0.003 in *C. virginica* and 0.002 in *A. islandica*. $K_B$ between seawater and the EPF is 0.8 in *C.*
*virginica* and 0.7 for *A. islandica* (Table 3).

| | Control<br>*A. islandica*<br>($\Omega_{aragonite}$) | Control<br>*C. virginica*<br>($\Omega_{calcite}$) | Moderate OA<br>*C. virginica*<br>($\Omega_{calcite}$) | High OA<br>*C. virginica*<br>($\Omega_{calcite}$) |
|---|---|---|---|---|
| $\Omega$ using EPF pH (range) | 1.7 (1.0-3.8) | 3.7 (1.3-11.4) | 1.1 (0.5-2) | 0.9 (0.5-1.2) |
| $\Omega$ using $\delta^{11}$B-calculated pH (range) | 3.8 (2.9-6.7) | 15.4 (6.7-37) | 6.1 (3-11.7) | 6.5 (3.4-9.7) |

Table 3. Table of calculated saturation state ($\Omega$) with respect to calcite (C. virginica) or aragonite (A. islandica) for the
average EPF pH value based on microelectrode measurements or δ11B-calculated EPF pH.




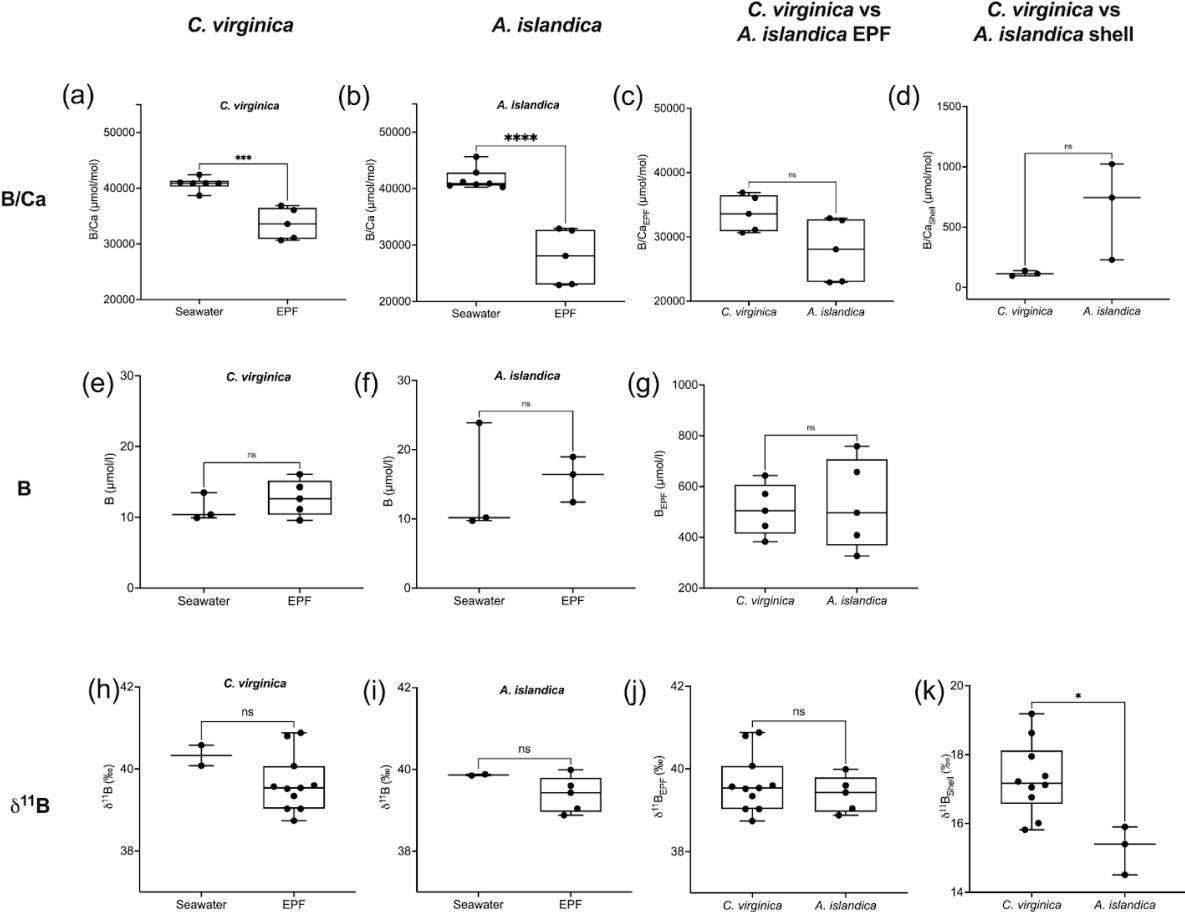

f 06

Figure 6. Box plots of B/Ca comparing seawater and extrapallial fluid for (a) C. virginica and (b) A. islandica, (c) comparing EPF B/Ca between species, and (d) shell B/Ca between species. Box plots of [B] comparing seawater and extrapallial fluid for (e) C. virginica and (f) A. islandica, (g) comparing EPF [B] between species. Box plots of 11B comparing seawater and extrapallial fluid for (h) C. virginica and (i) A. islandica, comparing EPF 11B between species, and (d) shell 11B between species. Stars denote statistically different means and 'ns' signify non significant mean differences in a pairwise t-test (at significance $p < 0.05$).

There was no significant difference in $\delta^{11}B$ between seawater and EPF for both species in the control condition (Fig 6h-l). There was also no significant difference in EPF $\delta^{11}B$ between species(Fig 6j); however, there was a significant difference in shell $\delta^{11}B$ between *C. virginica* and *A. islandica* (t-test, n1=10 n2=3, p-value<0.05, Fig 6k). Under control conditions, shell $\delta^{11}B$ was measured to be $15.26 \pm 0.41$‰ (2 SD, n=3) for *C. virginica* and $18.34 \pm 0.59$ ‰ (2 SD, n = 3) for *A. islandica*.



**3.4** *Crassostrea virginica* **ocean acidification experiment geochemistry**







f 07



Figure 7. Box plots showing C. virginica (a) EPF B/Ca and (b) shell B/Ca across seawater pH treatments. Additionally, box
plots show (c) EPF [B], (d) EPF [Ca], (e) EPF 11B, and (f) shell 11B. Stars denote statistically different means and 'ns'
signify non significant mean differences in a pairwise t-test (at significance p < 0.05). The sample set for (e) was limited and
we were unable to analyze the lowest pH treatment.
In the *C. virginica* acidification experiment, EPF B/Ca but not shell B/Ca was found to increase as seawater pH decreased
(ANOVA p-value<0.05, compare Fig 7a-b). The EPF but not shell B/Ca was found to increase as EPF pH decreased
(regression p-value< 0.05, Fig 8a-b). The boron concentration of the EPF, but not the shell, significantly decreased with
decreasing EPF pH (regression p-value< 0.05, Fig 8c). The EPF B concentration increased with increasing seawater pH
(ANOVA p-value< 0.05, Fig 8c); however, shell boron concentrations did not significantly change with seawater pH. Due to
small EPF sample volume, EPF for the oysters in the lowest seawater pH treatment was not measured for $\delta^{11}$B. There was a
significant difference in mean EPF $\delta^{11}$B between the control pH treatment which was 39.39 ‰ and moderate pH treatment
which was 38.92 ‰ (t-test, n1=11 n2=7, p-value<0.05, Fig 7e-f). The difference between seawater $\delta^{11}$B and EPF $\delta^{11}$B was
0.91 ‰ for the control treatment and decreased to 0.47 ‰ for the moderate pH treatment. Shell $\delta^{11}$B, but not EPF $\delta^{11}$B,
significantly decreased with decreasing EPF pH (regression p-value<0.05, Fig 8e-f).





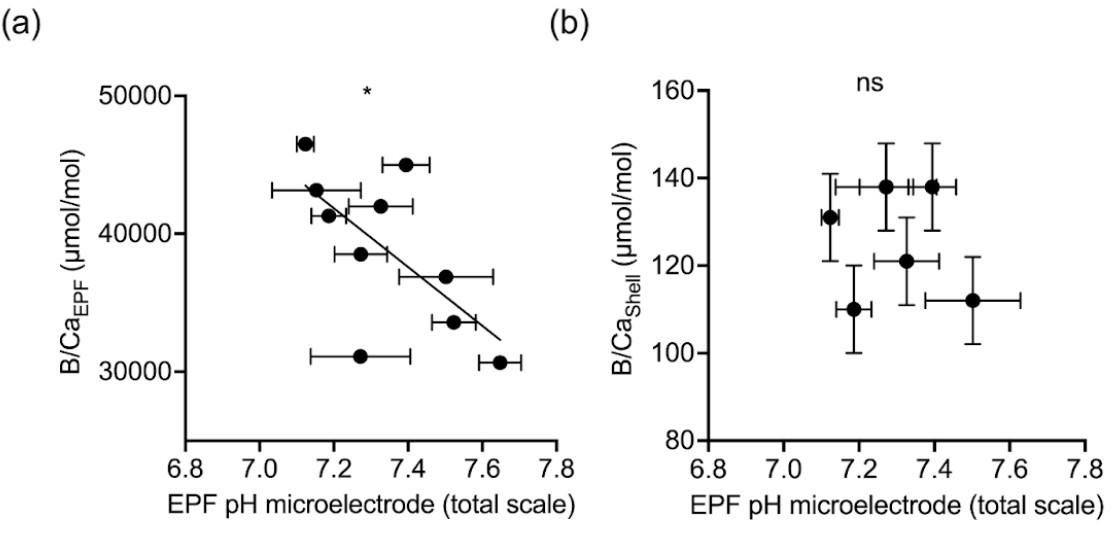

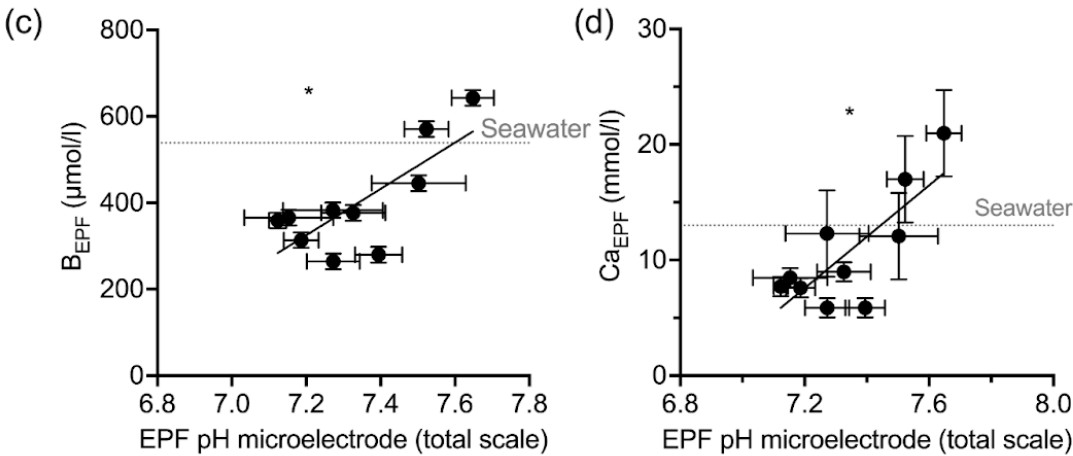

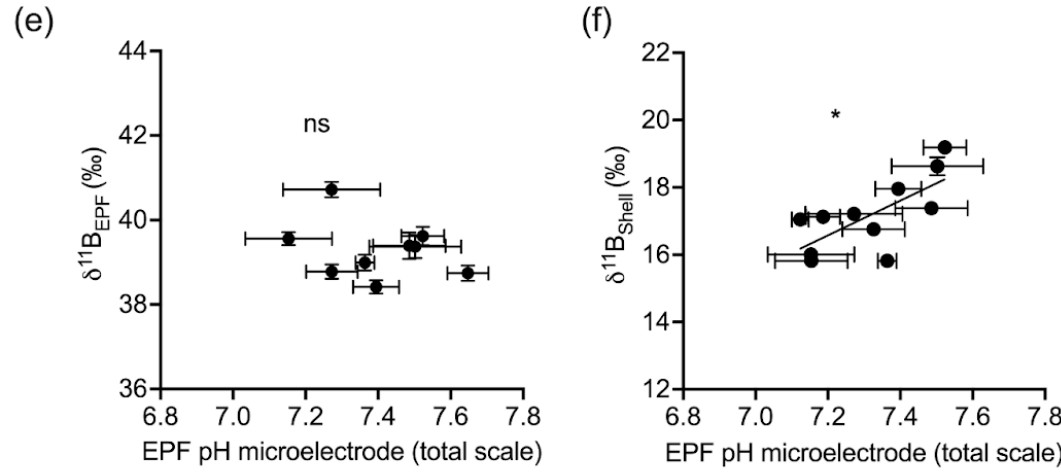

f 08



Figure 8. Scatter plots showing C. virginica individual specimen (a) EPF B/Ca and (b) shell B/Ca across corresponding
microelectrode EPF pH. Additionally, scatter plots of (c) EPF [B], (d) EPF [Ca], (e) EPF 11B, and (f) shell 11B across
microelectrode EPF pH. Stars denote statistically significantly nonzero regression slopes and 'ns' signify non significant
regressions (at significance p < 0.05). Dotted gray lines on (c) and (d) show the average [B] and [Ca] seawater
concentration, respectively.

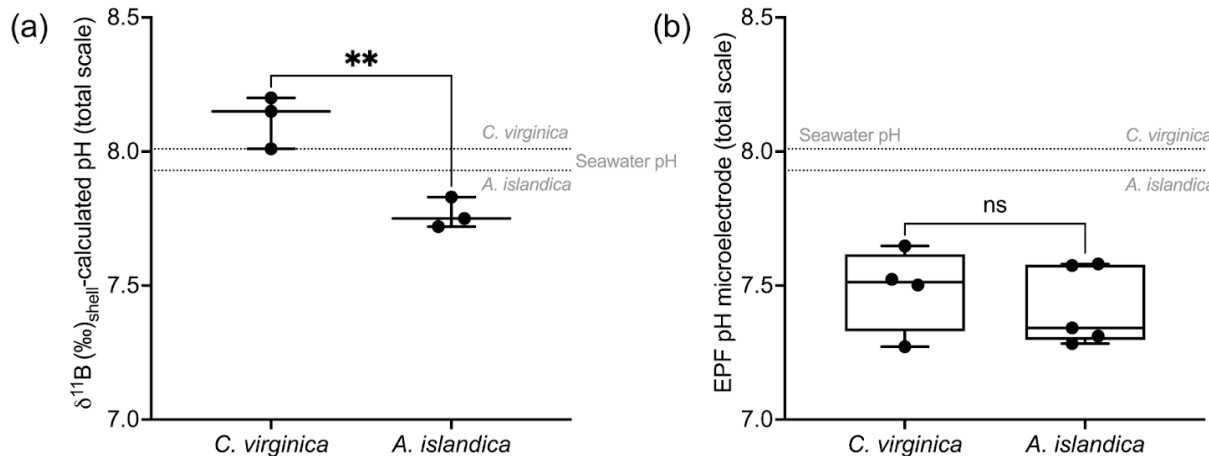

f 09


Figure 9. (a) Box plot of 11B-calculated pH for C. virginica and A. islandica. (b) Box plot of measured microelectrode pH
for C. virginica and A. islandica. The grey line shows seawater pH for C. virginica and A. islandica. Stars denote statistically
different means and 'ns' signify non significant mean differences in a pairwise t-test (at significance p < 0.05).
The control condition $\delta^{11}B$-calculated EPF pH for *C. virginica* was 8.12 ± 0.08 ‰ (2 SD, n=3) and for *A. islandica* was 7.93
± 0.09 ‰ (2 SD, n=3), which yielded a statistically significant difference between the two species (t-test, n1=3 n2=3,
p-value<0.05, Fig 9a). For *C. virginica*, the $\delta^{11}B$-calculated EPF was 0.1 pH units higher than the seawater pH and 0.6 lower
than measured EPF pH. Conversely, the *A. islandica* $\delta^{11}B$-calculated EPF was 0.1 pH units lower than the seawater pH and
0.3 higher than the measured EPF pH (Fig 9). Fig 10a shows the measured EPF pH, the $\delta^{11}B$-calculated EPF, and seawater to
EPF 1:1 pH line graphed across the *C. virginica* acidification experiment. The slope of the measured microelectrode EPF pH
versus seawater pH linear regression was 0.3, and lies below the seawater to EPF 1:1 pH line, but intersects the seawater to
EPF 1:1 pH line at lowest pH/highest $p$CO$_2$ culture conditions (Fig 10). Conversely, the slope of the $\delta^{11}B$-calculated EPF pH



versus seawater pH linear regression was 0.1, lies above the seawater to EPF 1:1 pH line, but intersected the seawater to EPF
1:1 pH line at higher culture pH conditions (Fig 10).

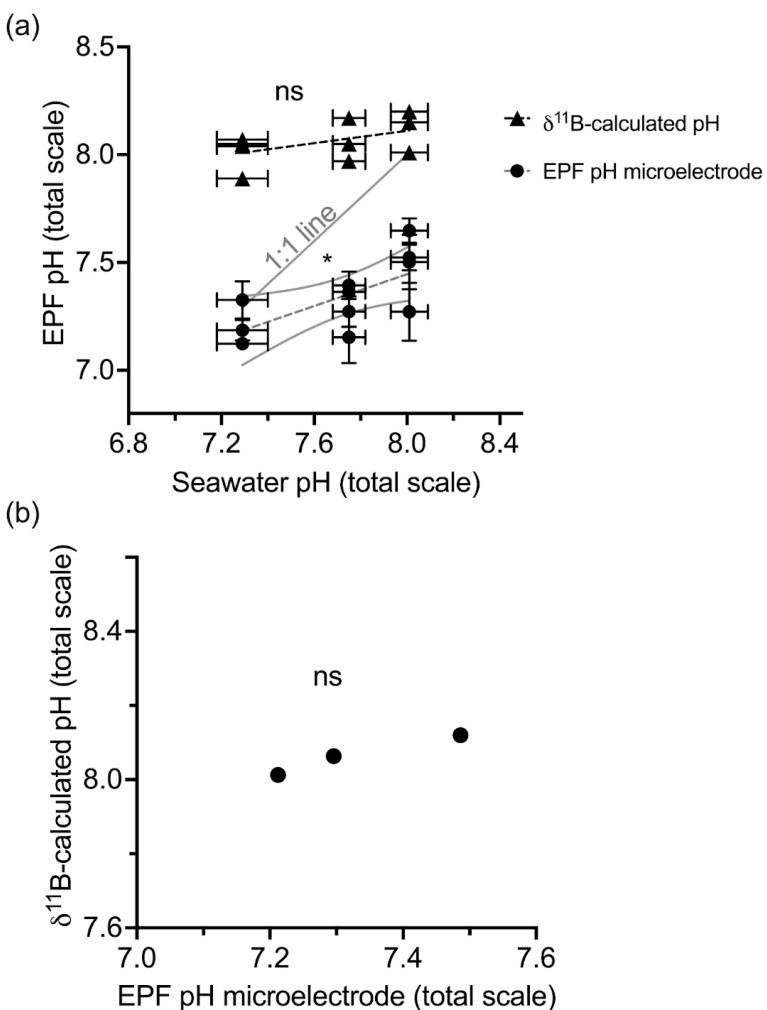

f 10


Figure 10. (a) Scatter plot of 11B-calculated pH and microelectrode EPF pH across seawater pH treatments. The gray line shows the 1:1 seawater to EPF pH line. In the seawater pH: EPF pH space, the 11B-calculated pH regression line is statistically nonzero (at significance $p < 0.05$), with a slope of 0.368. The microelectrode EPF pH line was not significantly nonzero and had a slope of 0.143. (b) shows the averaged 11B-calculated pH versus microelectrode EPF pH. Stars denote statistically significantly nonzero regression slopes and 'ns' signify non significant regressions (at significance $p < 0.05$).



For the *C. virginica* acidification experiment, Downey-Wall et al., (2020) measured the EPF pH of individual specimens in
each acidification treatment over a 24-hour period ($n_{total}$=108 and n=6 per time point per treatment). Fig 11 shows how the
EPF pH for each individual fluctuated over 24 hours. The control treatment EPF pH of individuals did intersect the averaged
seawater pH for the treatment tanks, however, the EPF pH in the moderate and high pH treatments fell below the
corresponding average treatment seawater pH lines. For all treatments, the time series EPF pH lines fell below the
corresponding treatment averaged $\delta^{11}$B-calculated EPF pH line.



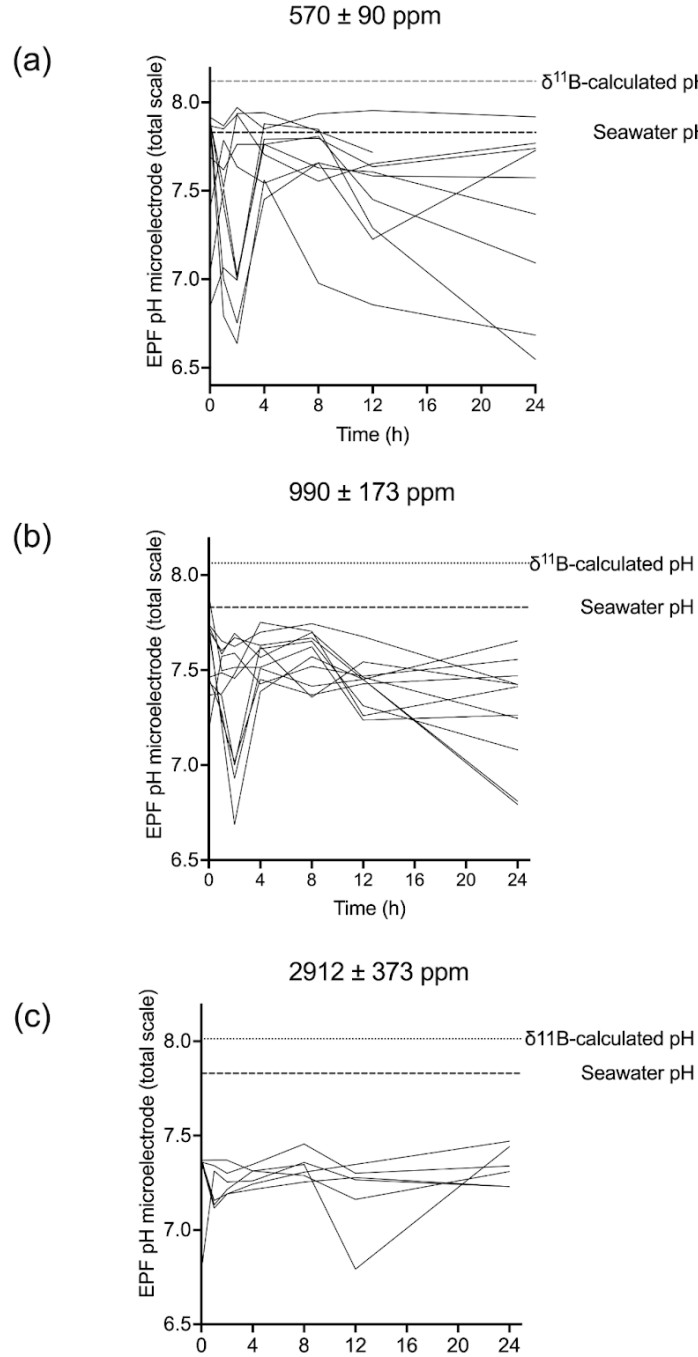

f 11


Figure 11. Time series (in hours) of microelectrode EPF pH over a 24 hour period for (a) control (b) moderate and (c) high
pCO2 treatments. Each line represents the microelectrode EPF pH for each individual specimen measured in that treatment.





The small dotted line shows the corresponding average 11B-calculated pH for the treatment and the larger dotted line shows
the average seawater pH for the treatment.
In Table 3, the EPF aragonite saturation state ($\Omega_{aragonite}$) for *A. islandica* and EPF calcite saturation state ($\Omega_{calcite}$) for *C.*
*virginica* were calculated using the averaged measured EPF pH and averaged $\delta^{11}B$-calculated EPF pH, averaged measured
$[Mg^{2+}]$ and $[Ca^{2+}]$, and literature values of DIC (3000 μmol/L for *A. islandica* taken from Stemmer et al. (2013) and 4200
μmol/L for *C. virginica* from McNally et al. (2022). Under control conditions, the *A. islandica* $\Omega_{aragonite}$ and *C. virginica* $\Omega_{calcite}$
that was calculated using $\delta^{11}B$-calculated EPF pH and measured EPF pH (Table 3). Under the ocean acidification
experiment, EPF $\Omega_{calcite}$ decreased with decreasing seawater pH when using either EPF pH or $\delta^{11}B$-calculated EPF pH to
calculate EPF $\Omega_{calcite}$. There were large differences in *A. islandica* $\Omega_{aragonite}$ and *C. virginica* $\Omega_{calcite}$ when using either EPF pH
($\Omega_{aragonite}$=1.7 and $\Omega_{calcite}$=3.7) or the $\delta^{11}B$-calculated pH ($\Omega_{aragonite}$=3.8 and $\Omega_{calcite}$=15.4).
**4. Discussion**
**4.1 $[Mg^{2+}]$ and $[Ca^{2+}]$ concentrations in the EPF and shell**
This study examined tripartite element and isotope fractionation between different reservoirs involved in the
biomineralization of two bivalves species, aragonitic *A. islandica* and calcitic *C. virginica*. Marine bivalves source ions for
internal fluids from seawater and previous studies by Crenshaw (1972) have highlighted that the extrapallial fluid, the
internal ion reservoir pool for calcification, is chemically different from seawater. Seawater enters the hemolymph fluid
within the bivalve tissues through the gills, filter feeding, and passive diffusion. Thereafter, the ions sourced from seawater
are modulated either passively or actively across the outer mantle epithelium (OME) cells into the extrapallial cavity, a
semi-isolated space that separates the outer mantle epithelium tissue from the shell. Here, ions are sourced to the site of
calcification where biomineralization occurs. The exact mechanisms behind bivalve biomineralization is still a topic of
active research and evidence has been put forth for several distinct pathways, primarily regulation of calcification
constituents across the OME and transport of a precursor phase of $CaCO_3$ to promote calcification (Addadi 2003; Checa

2020).

In the complementary study by Downey-Wall et al. (2020), it was found that the *C. virginica* calcification rates
decreased with seawater pH (Downey-Wall et al., 2020; Fig 2). The reduction of calcification under ocean acidification
conditions is well documented in other seawater pH experiments on different bivalve species (e.g., Ries et al., 2009; Beniash
et al., 2010; Waldbusser et al., 2011; Downey-Wall et al., 2020). This result is consequential as the shell is important in
protecting the animal from predation, desiccation, and the effects of transient changes in seawater chemistry (Gosling 2008).
Under ambient control conditions, *C. virginica* and *A. islandica* microelectrode EPF pH was lower than seawater pH.
Additionally, under both the moderate and high experimental ocean acidification treatments, the average microelectrode EPF
pH of *C. virginica* was lower than seawater pH. These findings are in line with previous work on bivalves, which show that



the EPF pH is regularly lower than seawater pH (Crenshaw 1972, Heinemann et al., 2012, Stemmer et al., 2013, Sutton et al.,
2018; Cameron et al. 2019, Liu et al., 2020) and that simulated ocean acidification results in a decreased EPF pH
(Michaelidis et al., 2005; Thomsen et al., 2013, Zittier et al., 2015, Cameron et al., 2019; Downey-Wall et al., 2020).
However, the change in pH between EPF and seawater pH ($\triangle$pH) decreased with decreasing pH, resulting in an EPF pH that
was closer to seawater pH under acidified conditions (Table 1).

Here we show that, under ambient conditions, both the EPF Mg/Ca and B/Ca of both *C. virginica* and *A. islandica*
were lower than that of seawater, indicating that the EPF has a distinct geochemical make up different from seawater (Fig 3;
Downey-Wall et. al., 2022). This is consistent with the anatomical understanding in bivalves that EPF is semi-isolated from
seawater and its geochemistry can be influenced by ion fluxes across the OME as well as other ion pathways (Crenshaw
1972; Stemmer et al., 2013; Sillanpaa et al., 2018). However, we also find that for both Mg/Ca and B/Ca, this result is driven
by an increase in absolute [$Ca^{2+}$] in EPF, so we do not find evidence for dilution or concentration of the absolute [$Mg^{2+}$] or
Bin the EPF (Fig 3). Previous work on bivalves has shown that magnesium can inhibit calcite crystal nucleation and there is
evidence for exclusion of [$Mg^{2+}$] from the EPF (Lorens and Bender, 1977). In line with other studies, we show that *C.*
*virginica* and *A. islandica* have lower Mg/Ca in EPF than seawater (Lorens and Bender, 1977; Planchon et al., 2013);
however, we note that the EPF Mg/Ca trend is driven by changes in EPF Ca. *C. virginica* and *A. islandica* EPF Mg/Ca were
significantly different, with lower EPF Mg/Ca for *A. islandica*, possibly due to different controls over EPF [$Ca^{2+}$] between
both species. The partition coefficient between EPF and the shell was calculated to be 0.003 for *C. virginica* 0.0002 for *A.*
*islandica*, which is consistent with previous studies on bivalves and with the Mg/Ca mineralogical difference between the
calcite produced by *C. virginica* and the aragonite produced by *A. islandica* (Ulrich et al. 2021).

We found that the EPF $\delta^{26}$Mg of *C. virginica* was depleted compared to seawater $\delta^{26}$Mg (Fig 3). Our $\delta^{26}$Mg values
for the EPF and shell were in line with previous work on bivalves (Planchon et al., 2013). Planchon et al. (2013) found a
-0.23 ± 0.25 ‰ (2 SD, n=5) difference between EPF and seawater in the aragonitic manila clam, *Ruditapes philippinarum*.
Similarly, in the present study, a difference of -0.11 ± 0.06 ‰ was observed for the calcitic *C. virginica*, but no $\delta^{26}$Mg data
were collected for *A. islandica* due to sample limitation. Both Planchon et al. (2013) and the present study show depleted
EPF $\delta^{26}$Mg relative to seawater $\delta^{26}$Mg, indicating a potential biological modulation of EPF [$Mg^{2+}$] which has been previously
attributed to heavier isotopes being incorporated into soft tissues or magnesium fixation within organic molecules (Planchon
et al., 2013). However, it is important to note that the difference between EPF and seawater $\delta^{26}$Mg is low and the $\delta^{26}$Mg
fractionation between the shell and seawater (2.43‰) was slightly larger than but still in line with inorganic calcite
precipitation studies (Mavromatis et al., 2013; Saulnier et al., 2012).

Only *C. virginica* was cultured under ocean acidification (OA) treatments representing control, moderate, and high
OA treatments. As mentioned above, the control experiment showed elevation of EPF [$Ca^{2+}$] and EPF [$Mg^{2+}$] relative to
seawater. However, as EPF pH decreased, the  EPF [$Ca^{2+}$] and [$Mg^{2+}$] significantly decreased as well (Fig 3 & 5). Ion
transporters such as voltage gated Ca-channels tend to also affect chemically similar ions like [$Mg^{2+}$] and a reduction of such
a transporter could possibly explain the similar trends in [$Ca^{2+}$] and [$Mg^{2+}$] concentrations under OA (Hess et al., 1986).



Under OA conditions, EPF [Ca²⁺] decreased to concentrations that were similar to or below seawater Ca, indicating a
reduced ability of the organism to upregulate these ions under OA conditions. Previous studies have found a similar tight
coupling between pH and Ca. For example, Stemmer et al. (2013) found synchronous patterns between pH and [Ca²⁺]
dynamics in *A. islandica* that they explained to be the result of calcium-transporting ATPase, which exchanges protons and
calcium ions across the OME and has proven to be important for acid-base regulation and calcium transport in bivalves
(Stemmer et al., 2013; Sillanpaa et al., 2018, 2020). Although calcium transporting ATPase could explain this increase in
[Ca²⁺] under ambient conditions, this transport mechanism may be reduced under acidified conditions, thereby impairing the
bivalve's ability to regulate protons and calcium ions in the extrapallial fluid, rendering EPF [Ca²⁺] and pH more similar to
that of seawater.
Alternatively, the simultaneous reduction in [Ca²⁺] and [Mg²⁺] under OA conditions could point to an ion storage
mechanism. The reduction of both calcium and magnesium within the EPF under moderate and high OA treatments could
possibly be linked to changes of storage and budgets of ions under stressful conditions (Mount 2004; Johnstone et al., 2015;
Wang et al. 2017). Further, several studies have highlighted significant changes in bivalve [Ca²⁺] ion transport and storage in
different extracellular and subcellular compartments associated with shell damage and repair under acidified conditions
(Sillanpaa et al., 2016; Mount et al., 2004; Fitzer et al., 2016). Lastly, the EPF [Ca²⁺] could simply reflect the balance
between calcification and dissolution of the shell, despite the decrease in calcification rate over the experimental period, as
exemplified by a study on *C. virginica* conducted by Ries et al. (2016) that found that under similarly low saturation states,
localized shell calcification was maintained despite net dissolution of the shell. Regardless of the exact mechanism, the
reduction in extrapallial fluid [Ca²⁺] under ocean acidification is a significant result that could impact the ability of bivalves
to calcify by decreasing the CaCO₃ saturation state of the EPF.

**4.2 Boron geochemistry**
The boron isotopes and B/Ca proxies have been used as paleo-pH and CO₃²⁻ proxies, respectively, recording
changes in seawater carbonate chemistry in the shells of foraminifera (Hemming and Hanson 1992; Sanyal et al., 2001;
Foster and Rae 2016). In corals, however, there is evidence that these proxies monitor changes in the carbonate chemistry of
the internal calcifying fluid, which may be different from seawater geochemistry (Allison and Finch 2010; Sutton et al.,
2018; Guillermic et al., 2021). The boron isotopes proxy has also been applied to other marine species (Sutton et al., 2018,
Liu et al., 2020, Cornwall et al., 2017), but independent measurements are needed to fully understand the systematics of this
proxy in other organisms. In the present study, we constrained the B/Ca and δ¹¹B of the main reservoirs involved in the
biomineralization (seawater, extrapallial fluid, and shell) of two species of bivalves, the oyster *C. virginica* and the clam *A.*
*islandica*.
For both *A. islandica* and *C. virginica*, there were no significant changes nor correlation observed between δ¹¹B of
the EPF and seawater (Fig 6). Shell δ¹¹B was significantly different between species, with *A. islandica* recording lower shell
δ¹¹B (15.26 ± 0.41 ‰) than *C. virginica* (18.34 ± 0.59 ‰). Using boron isotope systematics, the δ¹¹B-based EPF pH was





determined to be 7.76 ± 0.07 for *A. islandica* and 8.12 ± 0.09 for *C. virginica*. The $\delta^{11}$B-based pH was significantly different
between the two species (t-test p value <0.05) and also significantly different from the direct EPF microelectrode pH
measurements of 7.41 ± 0.14 and 7.48 ± 0.15 for *A. islandica* and *C. virginica*, respectively (t-test p value < 0.05). In other
words, the use of canonical $\delta^{11}$B proxy systematics to calculate $\delta^{11}$B based pH does not match direct measurements of EPF
pH. Microelectrode EPF pH was consistently lower than seawater for both species. $\delta^{11}$B-based pH also revealed EPF pH
lower than seawater pH for *A. islandica* (but to a lesser extent than direct microelectrode measurement), but an EPF pH
greater than seawater for *C. virginica*. This observation in the control experiments holds true under ocean acidification,
where the $\delta^{11}$B-based pH is systematically higher than microelectrode EPF pH (Fig 10). Both $\delta^{11}$B-based pH and measured
EPF pH record a decrease in pH under acidified conditions (regression p<0.05 for microelectrode pH). However, the offset
between microelectrode EPF pH and the $\delta^{11}$B-calculated pH was 0.3 pH units and increased to 0.6 and 0.8 pH units for the
moderate and high OA treatments, respectively (Table 1). This demonstrates that, under OA conditions, the incongruence
between $\delta^{11}$B based pH and measured EPF pH increases and potentially renders the seawater pH proxy impractical, even
after species-specific empirical calibration. Shell $\delta^{11}$B was not correlated with seawater pH, but was significantly correlated
to microelectrode pH. These data indicate that microelectrode EPF pH does not fully resolve $\delta^{11}$B vital effects. However it is
important to note the differences in timescales associated with $\delta^{11}$B-calculated EPF pH and microelectrode pH. Our
microelectrode pH measurements, although averaged across several time points, show snapshots in time and is variable due
different behavioral scenarios such asn open (feeding, high pH) and closed (respiring into a closed system, low pH) cycles.
Conversely, the $\delta^{11}$B approach represents EPF pH integrated average EPF pH over the interval that the sampled shell was
formed, which could range from days to weeks. Furthermore, the $\delta^{11}$B method will only record EPF pH when the shell is
forming, which can skew the archiving of the $\delta^{11}$B (pH) signal in the shell to higher values because the crystal only forms
when saturation states and calcification rates are higher. This potential bias is also consistent with our $\delta^{11}$B-calculated EPF
pH data being higher than the microelectrode pH data, and similar to trends seen in the corals (Cameron et al, 2022).

A possible explanation for the incongruence between $\delta^{11}$B-based pH and measured EPF pH arises from boron

isotope systematics. The boron isotope proxy assumes that only the charged borate ion is incorporated as $BO_4$ into the
mineral but has been shown that boric acid can also be incorporated as $BO_3$, and NMR studies have shown the presence of
$BO_3$ in the shells of different marine organisms (Rollion Bard et al., 2011; Cusack et al., 2015). However, the presence of
$BO_3$ does not obviously translate to a strong bias in the $\delta^{11}$B signature of the mineral due to the potential re-coordination of
$BO_4$ to $BO_3$ within the crystal lattice (Klochko et al., 2009). A simple calculation shows that 14-17% boric acid incorporation
could explain the observed difference between EPF pH and $\delta^{11}$B-calculated pH for *C. virginica*, with only 6% boric acid
incorporation needed for *A. islandica*, which could very well explain the discrepancy. Alternatively, shell $\delta^{11}$B could also be
affected by seawater or extrapallial fluid DIC, which bivalves are known to modulate under ambient and OA conditions
(Crenshaw 1972, Stemmer et al., 2012). Gagnon et al. (2021) found that the shell $\delta^{11}$B of deep-water coral is independently
sensitive to changes in seawater DIC as a result of diffusion of boric acid (Gagnon et al., 2021), though no similar studies
have looked at the same effect in bivalves this mechanism is still possible. Taken together, these findings could explain the




offset between δ¹¹B-based pH and seawater or EPF pH. Nevertheless, this remains speculative as there is no further evidence
of boric acid incorporation in these species.

Furthermore, boron isotope derived pH can be influenced by diffusion of boric acid across cell membranes (Stoll et

al., 2012; Liu et al., 2018; Liu et al., 2021; Gagnon et al., 2021). At two extremes, diffusion between seawater and the
calcifying fluid pool can be fast, resulting in chemically and isotopic equilibrium between both pools, or diffusion can be
slow, resulting in calcifying fluid being isolated from seawater such that the boron isotopes would record the chemistry of
the calcifying fluid under physiological control. If diffusion is fast compared to other processes, then seawater and the
calcifying fluid would be in equilibrium and the δ¹¹B would not differ between the two pools. Our data show no difference
between seawater and EPF δ¹¹B. However, differences in Ca, Mg, and δ²⁶Mg between seawater and EPF does provide
evidence for physiological modulation of the EPF, despite similar δ¹¹B signatures.

In the case where there is not a strong diffusion of boric acid, then the pH calculated from boron isotopes should

reflect the pH at the site of calcification and physiological control over the calcifying fluid. The difference between
microelectrode EPF pH and δ¹¹B-based EPF pH implies that pH measured with boron isotopes probes a localized site of
calcification rather than the entire EPF pool measured with microelectrode. A spatial and temporal study conducted by
Stemmer et al. (2019) measured the EPF of *Arctica islandica* and showed highly dynamic changes in pH, [Ca²⁺] and DIC
from the surface of the shell to the outer mantle epithelium (OME), with localized environment at the OME reaching pH
values up to 9.5. Due to this high variability, it is possible that the EPF microelectrode measurements in this study did not
capture the full variability of the EPF. Stemmer et al. (2019) presented EPF pH values measured at the shell surface ranging
[7.1-7.6] for *A. islandica*, comparable to the values measured from microelectrode in this study. Additionally, Stemmer et al.
(2019) found large influxes of DIC which could not have been explained just from metabolic activity, but instead indicated
intense DIC pumping and bursts of calcification. These findings are in line with the holistic view of biomineralization
outlined in Checa (2018) and Johnstone (2015) that argue that crystal deposition is a series of periodic events under
biological regulation. In our study, a time-series of microelectrode EPF pH shows that at no point, during ventilation and
closed cycles, does the EPF pH reach the δ¹¹B-calculated pH (Fig 11). The fact that microelectrode EPF pH is systematically
lower than seawater pH for both of our bivalve species may reflect localized differences in pH associated with zones of
calcification. The two environments (site of calcification and bulk EPF) can act distinctly, with low pH and high DIC EPF
being a source of carbon for the site of calcification, and with the elevated pH of the site of calcification supporting the
conversion of the DIC species to [CO₃²⁻] in support of mineral precipitation. Further work would be needed to assess this
highly dynamic and localized environment, however our study shows that boron isotopes may reflect the pH of the
microenvironment where calcification occurs within the EPF, which has previously been inferred by prior studies using
non-geochemical approaches (Ramesh et al., 2017; Stemmer et al., 2019).



## Conclusion

In this study, we used numerous approaches constraining the geochemical composition of and partitioning between the tripartite reservoirs of bivalve mineralization system--seawater, the EPF and the shell. Our study presents Mg/Ca and B/Ca, and absolute [$Ca^{2+}$] data of the seawater, EPF and shell. Comparisons of seawater and extrapallial fluid Mg/Ca and B/Ca, Ca, and $\delta^{26}Mg$ indicate that the EPF has a distinct composition that differs from seawater. Additionally, our OA experiments show that the EPF Mg/Ca and B/Ca, as well as absolute Mg, B, and Ca, all were significantly affected by $CO_2$-induced ocean acidification, demonstrating that the biological pathways regulating or storing these ions involved in calcification are impacted by ocean acidification. Decreased calcium ion concentration within the extrapallial fluid due to OA could impair calcification by lowering the saturation state of the EPF with respect to $CaCO_3$. Additionally, our results show that shell $\delta^{11}B$ does not faithfully record seawater pH. However, shell $\delta^{11}B$ is correlated with EPF pH, despite an offset from *in situ* microelectrode pH measurements. Both microelectrode pH and $\delta^{11}B$-calculated pH decreased with decreasing pH. However, the $\delta^{11}B$-calculated pH values were consistently higher than microelectrode pH measurements, indicating that the shell $\delta^{11}B$ may reflect pH at a more localized site of calcification, rather than pH of the bulk EPF. Furthermore, the offset between the $\delta^{11}B$-calculated pH and microelectrode pH increased with decreasing pH under ocean acidification, indicating OA has a larger effect on bulk pH of the EPF measured via microelectrode than on site of calcification pH—the latter of which the bivalve may have more physiological control over to ensure continued calcification, even under chemically unfavorable conditions. These complex dynamics of EPF chemistry suggest that boron proxies in these two bivalve species are not straightforwardly related to seawater pH, precluding utilization of those species for reconstructing the carbonate chemistry of seawater. Moreover, the $\delta^{11}B$ proxy may not be suitable for reconstructing seawater pH for bivalves with high physiological control over their internal calcifying fluid and is further complicated under conditions of moderate and extreme ocean acidification, where $\delta^{11}B$ EPF pH deviates further from bulk microelectrode pH, possibly due to the effect of DIC on shell $\delta^{11}B$ or the tendancy for shell $\delta^{11}B$ to reflect EPF pH at the more localized site of calcification, rather than pH of the bulk EPF.

## Author contribution

LPC, AD, JBR, and KL designed the experiments and carried them out. BAC, MG, and RAE developed the geochemical study. BAC and Mg performed geochemical analysis with the help of JNS and JAH. BAC, MG, and RAE prepared the manuscript with contributions from all co-authors.

## Competing interests

The authors declare that they have no conflict of interest.



## Acknowledgements

BAC was supported by the National Science Foundation Graduate Research Fellowship Program under Grant No. DGE-2034835 and the UC Eugene Cota-Robles Fellowship. BAC, MG, and RAE are supported by the Ocean Science work of Center for Diverse Leadership in Science which is funded by a grant from the David and Lucile Packard Foundation (no. 85180), National Science Foundation grant NSF-RISE-2024426, and by gifts from Oceankind and Dalio Philanthropies. The Center for Diverse Leadership in Science is also supported by NSF-RISE-2228198, the Waverly Foundation, the Silicon Valley Community Foundation, and the Sloan Foundation. KL and JBR were supported by the National Science Foundation grant BIO-OCE 1635423. The authors would like to thank Celine Liorzou, Yoan Germain, and Anne Trinquier for their technical support at the PSO. Additionally, the authors would like to thank Stefania Gili for her technical support at Princeton University.

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
