# Peer review of "Magnesium (Mg/Ca, $\delta^{26}\text{Mg}$ ), boron (B/Ca, $\delta^{11}\text{B}$ ), and calcium ( $\text{Ca}^{2+}$ ) geochemistry of *Arctica islandica* and *Crassostrea virginica* extrapallial fluid and shell under ocean acidification"

_EGUsphere, 2024_

## Author Comment (AC2)

RC1

This manuscript deals with an interesting topic: bivalve calcification processes in relation to ocean acidification. However, in a present for, it is not suitable for publication.

The introduction section is too long and should be significantly reduced. Parts of it are hard to follow. The Material and Methods are not sufficiently clearly presented, as relevant information is missing. The choice of target species is not clearly presented; it is not clear why the authors chose to analyse slow-growing Arctica islandica rather than some faster-growing species with an aragonitic shell. Shell sizes used, as well as the number of shells used in the research, are not clearly presented.

Reponse #1: In response to the constructive criticism regarding the general manuscript structure we propose a number of changes. 1. Reduce the introduction and edit it for clarity. 2. In the materials and methods section we can add requested information including that contained in the supporting publication (i.e. the number of samples for each measurement, sampling process, addition of methods on the statistical approach).

Regarding experimental design we studied one calcitic species (*C. virginica*) that has ecological and economic importance, and is used in paleoclimate research. The aragonitic species whilst slow growing has been the subject of numerous geochemical and also metagenomic studies developing it as an archive of environmental information (eg. ; Schöne et al., 2005; Milano et al., 2017; Liu et al., 2015; Der Sarkissian et al., 2017). Importantly these specimens have a lot of supporting information generated and published in a prior study - for example the carbonate chemistry of the pallial fluid has been well-characterized (Downey-Wall, et al., 2022).

Overall, the manuscript appears more as a draft version rather than a complete document. There are serious issues with the organisation of text, Figures and Tables. Numerous mistakes in text and references are questioning the systematic approach of authors in data analysis. Statistical analysis used is not fully credible. The number of Figures is too large. The discussion as well as Conclusions sections are too long. There is a certain level of repetition between the Discussion and Introduction sections, which is redundant and needs to be revised. In the Discussion section, authors need to start by clearly identifying their most important findings and explaining them in a clear and focused way in relation to previous research.

Reponse #2: In our revised manuscript we will address issues regarding the organization of the text, figures, and tables. We will add a subsection for statistical analysis in the materials and methods section to be more complete. Additionally, our new draft will be edited to decrease the number of figures. We will also edit the discussion and conclusion section to shorten and remove repetition.

To summarize our statistical approach for t-tests and ANOVA tests was:

Step A. Run a Shaprio-Wilks test to determine normality of residuals.

Step B. Transform non-normally distributed data and re-run Shapiro-Wilks test.
Step C. Run ANOVA with pH as a four level factor. ANOVA and t-test significance was achieved if the p-value was less than 0.05.
Regression analysis was performed using the GraphPad Prism software and significance was denoted if the slope of the regression was statistically non-zero.
We will write a revised version in which these methods are more clearly explained in the methods section.

Specific comments:

Line 38 – insert "e.g.," in the beginning of brackets. Use this approach in other parts of the text where you are just presenting selected references for a certain statement.
Line 43 – replace "e.g.." with "e.g.,"
Response #3 We can address these minor comments in the revision.

Line 43 & 50 & 137 – not sure what you mean by "reviewed", why not just citing the reference
Response #4 We changed the text to cite the reference instead of writing "reviewed by".
Line 60 – place relevant references after mentioning individual species names and not at the end of the sentence.
Response #5 We can address these minor comments in the revision.
Line 77 – did you mean Zhao et al. 2018? There is no publication from 2016 in reference list. Needs to be checked in detail.
Response #6 The correct reference was Zhao et al. 2017 which was added to the reference list.
Line 81 – a reference to the Figure should be before the Figure, move the Figure below this text
Response #7 We can rearrange to put the figure after the reference. We will make sure that all figures will be after it is cited in the text.
Figure 1 – consider increasing a bit font site on the bottom of the right figure part
Line 88 – remove extra dot
Line 89 – check the presentation of isotope names and formatting of numbers
Lines 88-90 Species names should be presented in italic font
Response #8 We can address these minor comments in the revision.
Table 1 – the title should be presented above the table. Extra horizontal and vertical lines should be omitted from the Table for clarity. This table should not be presented in the Introduction section – it actually contains the Results of this study. It is not sufficiently clearly explained why parameters marked with n.d. were not measured.
Response #9 We can add that the data could not be measured, which was explained in the methods section.
Line 96 – it is not clear why the authors refer here to Table 1 and Zhao et al. 2018 paper – and to which of them – as two are listed in the reference section. Did you want to refer to Figure 1? There you cite Zhao et al. 2016 paper.
Response #10 We can just refer to the figure in the revision.
Line 109 – At the end there is only a reference to Crenshaw (1972), so check "et al."
Line 111 – year missing after Crenshaw
Line 143- 145 place references after corresponding taxa, not at the end of the sentence

Response #11 We can address these minor comments in the revision.

Lines 150-160 & other parts of the text. There is no need to go in so much detail in relation to other taxa. You text is too wide and too descriptive. You need to focus on target taxa and main questions.

Response #12 As we shorten the introduction we will focus our text on our taxa or species of interest.

Line 164 Full species name is mentioned before in text, so abbreviations should be used here-

Response #13 We can use the abbreviation after writing the full species name.

Line 176 Authors need to provide a short description of collection and culturing in this manuscript, it is not sufficient to refer to earlier manuscript. There is no sufficient indication of year research was conducted.

Response #14 We will write a short description in our revision that will include the dates of the collection and experiment dates.

Line 178 & other parts of the text – remove "psu"

Response #15 We can address these minor comments in the revision.

Line 182 you need to clearly specify size of bivalves used in your research as well as sample number

Response #16 This information will be added to the revised methods and materials section.

Line 183 – check the formatting of geo coordinates

Response #17 We can address these minor comments in the revision.

Line 190 what do you ean by "see" here?

Response #18 "see" was removed from this reference.

Line 191 – here you refer to Table a that is presented several pages before – this is not appropriate approach to organisation of the manuscript

Response #19 We will address this comment in the restructuring of the manuscript.

Line 194 you need to define "dry weight". What did you measure & with what? What was the precission?

Line 195 You need to define "buoyant weight"

Response #20 We can add a more specific explanation of buoyant weight and technique and calculations in the revision.

Line 207 Arctica islandica is a slow growing species – you need to explain why you choose 14 days as period here.

Response #21 A better explanation can be provided in the methods section. We were interested in the geochemistry of the shell and EPF of *A. islandica* specimens under ambient $CO_2$ conditions, which these organisms naturally grew in. Our need to maintain specimens under laboratory conditions was specifically to examine the EPF fluid and sample while in controlled conditions.

Line 212 You need to present Figure/diagram clearly illustrating how sampling of shell carbonate material was conducted.
Response #21 We can add a figure to show how EPF sampling and geochemical sampling was conducted.

Line 232 A sentence should not start with a number, revise
Response #22 We can address this minor comment in the revision.

Line 289 You refer here to Table 3 – and it is presented much later in the manuscript. There are serious issues with the organisation of your text, Figures and Tables.
Line 294 & other parts of text - Species names should be presented in italic font.
Line 304 – check the formatting of Fig – "a" vs. "A"
Response #23 We can address these minor comments in the revision.

Line 320 – statistical analysis used should be clearly presented in Material and Methods section. Did you test your data for homogeneity of variance (requirement of t test)? See for example Fig 3f – variances are not clearly not homogeneous according to your data presentation.

Response #24 Addressed in response #2.

Careful formatting of references is needed. Journal names are not consistently presented, as some are referred to by full names and others by their abbreviations. I did not check all in detail, and the following are just some observations: Lines 644-647 – check for use of capital letters in publication titles. Line 646 replace "mollusc" with "Mollusc". Line 652 – place "Porites" in italics, also check other references and present all species scientific names in italics font. Line 670 – year should be placed at the end of the reference for consistency. Line 677 – doi missing. Line 681 – capital letters are used, needs to be corrected.

Response #25 We will carefully edit the references section which has a few organizational issues and lack of consistency.

References cited in response:
1. Schöne, B. R., Fiebig, J., Pfeiffer, M., Gleß, R., Hickson, J., Johnson, A. L., ... & Oschmann, W. (2005). Climate records from a bivalved Methuselah (Arctica islandica, Mollusca; Iceland). Palaeogeography, Palaeoclimatology, Palaeoecology, 228(1-2), 130-148.
2. Milano, S., Nehrke, G., Wanamaker Jr, A. D., Ballesta-Artero, I., Brey, T., & Schöne, B. R. (2017). The effects of environment on Arctica islandica shell formation and architecture. Biogeosciences, 14(6), 1577-1591.

3. Liu, Y. W., Aciego, S. M., & Wanamaker Jr, A. D. (2015). Environmental controls on the boron and strontium isotopic composition of aragonite shell material of cultured Arctica islandica. Biogeosciences, 12(11), 3351-3368.
4. Der Sarkissian, C., Pichereau, V., Dupont, C., Ilsøe, P. C., Perrigault, M., Butler, P., ... & Orlando, L. (2017). Ancient DNA analysis identifies marine mollusc shells as new metagenomic archives of the past. Molecular Ecology Resources, 17(5), 835-853.
5. Downey-Wall, A. M., Cameron, L. P., Ford, B. M., McNally, E. M., Venkataraman, Y. R., Roberts, S. B., ... & Lotterhos, K. E. (2020). Ocean acidification induces subtle shifts in gene expression and DNA methylation in mantle tissue of the Eastern oyster (Crassostrea virginica). Frontiers in Marine Science, 7, 566419.
6. Zhao, L., Schöne, B. R., & Mertz-Kraus, R. (2017). Delineating the role of calcium in shell formation and elemental composition of Corbicula fluminea (Bivalvia). Hydrobiologia, 790, 259-272.

R2

Response #1 We thank reviewer 2 for their opinion that our work represents a major advance in biomineralogy research. We go on to address their specific comments below.

The temperatures for A. islandica are worryingly warm, near the upper range of this clam's thermal tolerance (lines 179/180; Seawater was maintained at a pH of 7.93 ± 0.09, temperature of 18.2 ± 1 o C, and salinity of 35 psu for the aragonitic clam A. islandica in the control conditions  (Downey-Wall et al., 2020). The authors need to provide this as a caveat to the results. In other words, are the results scalable for all temperature ranges?? The authors should consider the results of Liu et al. (2015) Environmental controls on the boron and strontium isotopic composition of aragonite shell material of cultured Arctica islandica, Biogeosciences, 12, 3351-3368, doi:10.5194/bg-12-3351-2015, whereby there seemed to be a potential influence of warmer temperatures on boron isotopes.

Response #2 Thank you for this point. *A islandica* and *C. virginica* specimens were maintained at different temperatures (9°C and 18°C, respectively) ; this will be edited in the revised manuscript.

Add the length of time for the experimental calibration for both species in line 180 at the end. Response #3 We can add the length of time in the revision.

What are the ages and shell heights for the A. islandica shells? They grow very slowly, thus this is important to have these metrics in this study- (not just citing Downey-Wall et al. (2020))

"2.2 Calcification rate measurements Net calcification rate was calculated using the dry weight at the start and end of the experiment. Initial dry weight was measured at the start of exposure, on day 33 or 34, after the acclimation period (Downy-Wall et al., 2020). The buoyant weight was measured on either day 50 or 80 and the final dry weight was derived using a linear relationship between oyster dry weight and oyster buoyant weight (Ries et al., 2009)."

This may be suitable for juvenile mollusks but not for adults, especially A. islandica. What are the uncertainties in such measurements for large adult clams?

Why haven't the authors reported calcification rates for A. islandica. This is a central variable that needs to be considered (like Fig. 2a for oysters).

Response #4 We will clarify that calcification measurements were only conducted on *C. virginica* specimen, not *A. islandica*. Our calcification measurements were taken to understand how calcification was affected by ocean acidification treatments, which *A. islandica* were not exposed to.
Shell sizes can be added to the revision of the materials and methods section and a better explanation can be provided. We were interested in the geochemistry of the shell and EPF of *A. islandica* specimens under ambient $CO_2$ conditions. The *A. islandica* specimen grew under ambient $CO_2$ conditions as they were collected in their natural environment. Our need to maintain specimens under laboratory conditions was specifically to examine the EPF fluid and sample while in controlled conditions.
The longer term experiment for *C. virginica* was needed to have the faster growing *C. virginica* which would have laid down new growth under $CO_2$ treatments.

How are the authors confident that they sampled ONLY calcium carbonate reflecting the experiment? Did they stain the shells with calcein? Did they measure linear growth? This is most relevant for A. islandica because of relatively slow growth rates (i.e., see Liu et al., (2023) Resistant calcification responses of Arctica islandica clams under ocean acidification conditions, Journal of Experimental Marine Biology and Ecology, https://doi.org/10.1016/j.jembe.2022.151855.)

Response #5: We sampled a thin layer across the inner surface at the base of the shell, avoiding any parts such as repaired shell laid down to cover bore holes, as we thought the chemistry of those regions might be different. We don't have info such as a calcein stain to show

where the new growth is, so we sampled shell material that was in close contact with the EPF and was recently laid down. But we acknowledge that can be a potential bias.

Shell sampling – the organic matrix in shells contain about a magnitude more boron than in the shell, and this likely has a very different isotopic composition (value). Are the authors confident all organics were removed?

Response #6 Yes, before isotopic analysis carbonate samples went through two oxidative cleaning steps to remove organics from the carbonate material. The oxidative agent consists of H2O2  which is similar to other studies (Pre-treatment effects on coral skeletal δ13C and δ18O - ScienceDirect). This cleaning was performed on various marine organisms and led to consistent data within species (corals, coralline algae, oyster, Liu et al. 2020). We can add a sentence before line 233 to explain this in more detail rather than just stating " 2.5-3.0 mg of oxidatively cleaned shell powders were dissolved in 1N HCl."

Why are the authors explain how they sampled the oyster shells but not the clam shells? The methods should have a parallel structure.

Response #7 The sampling was performed in the same way for both bivalve species, this can be added into our methods section for clarity.

Very interesting result in Figure 3- showing different chemical composition of EPF compared to ambient seawater. Important finding that lots of folks have been suggesting but without the EPF evidence. And when you go to the shells even less Mg than seawater, and EPF. Thus the mollusks must be regulating calcifying fluids.

 I really think the authors are missing an opportunity by not exploring changes in the shell geochemistry from both species here with growth rates, shell height, age, etc. The applicability/scalability of the study is far less without the inclusion of such metrics. Why not include these data?

 Response #8 Some of those data were not collected for both species. We have calcification data for *C. virginica* however calcification was not recorded for *A. islandica*. With the data we do have, we can re-explore those metrics and add them to the revised manuscript.

Ok- now some praise for the authors:

This is an important study with important implications. We learned that oysters (C. virginica) and clams (A. islandica) incorporate some elements and boron isotopes differently. The boron isotopic composition of the EPF for both species is different than seawater. The breakthrough of being able to sample the EPF chemistry/pH is a major advance in biomineralogy. Thus, a mechanistic model for biomineralization can be advance. Also, the mollusks evaluated here are not simple pH meters, and the shell d11B value is a mixture of the seawater d11B value and physiology. These results are consistent with in prep work that I am aware of now. Despite some issues with the description of the experiment and other concerns noted above, this is a major advancement.

References cited in response:
1. Grottoli, A. G., Rodrigues, L. J., Matthews, K. A., Palardy, J. E., & Gibb, O. T. (2005). Pre-treatment effects on coral skeletal δ13C and δ18O. Chemical Geology, 221(3-4), 225-242.
2. Liu, Y. W., Sutton, J. N., Ries, J. B., & Eagle, R. A. (2020). Regulation of calcification site pH is a polyphyletic but not always governing response to ocean acidification. Science advances, 6(5).

---

## Author Response (AR1)

RC1

This manuscript deals with an interesting topic: bivalve calcification processes in relation to ocean acidification. However, in a present for, it is not suitable for publication.

The introduction section is too long and should be significantly reduced. Parts of it are hard to follow. The Material and Methods are not sufficiently clearly presented, as relevant information is missing. The choice of target species is not clearly presented; it is not clear why the authors chose to analyse slow-growing Arctica islandica rather than some faster-growing species with an aragonitic shell. Shell sizes used, as well as the number of shells used in the research, are not clearly presented. Overall, the manuscript appears more as a draft version rather than a complete document. There are serious issues with the organisation of text, Figures and Tables. Numerous mistakes in text and references are questioning the systematic approach of authors in data analysis. Statistical analysis used is not fully credible. The number of Figures is too large. The discussion as well as Conclusions sections are too long. There is a certain level of repetition between the Discussion and Introduction sections, which is redundant and needs to be revised. In the Discussion section, authors need to start by clearly identifying their most important findings and explaining them in a clear and focused way in relation to previous research.

Overall, the manuscript appears more as a draft version rather than a complete document. There are serious issues with the organisation of text, Figures and Tables. Numerous mistakes in text and references are questioning the systematic approach of authors in data analysis. Statistical analysis used is not fully credible. The number of Figures is too large. The discussion as well as Conclusions sections are too long. There is a certain level of repetition between the Discussion and Introduction sections, which is redundant and needs to be revised. In the Discussion section, authors need to start by clearly identifying their most important findings and explaining them in a clear and focused way in relation to previous research.

Reponse #1: In response to the constructive criticism regarding the general manuscript structure we changed the following. 1.  We reduced the introduction substantially, edited for clarity, and removed redundancy. The word count was reduced from 2213 to 1256 words. 2.  In the materials and methods section we added the requested information including that contained in the supporting publication (i.e. the number of samples for each measurement, sampling process, addition of methods on the statistical approach). 3. We also edited the discussion section to shorten and remove repetition.

In our revised manuscript we addressed issues regarding the organization of the text, figures, and tables. Tables and figures were placed after the first text reference. Two figures that had redundant information were removed from the main text and will be placed in the supplemental materials section. The layout of the remaining figures were renumbered. We added a subsection for statistical analysis in the materials and methods section.

To summarize our statistical approach for t-tests and ANOVA tests:

Step A. Run a Shaprio-Wilks test to determine normality of residuals and a Brown-Forsythe test was used to determine heterogeneity of variance of residuals.

Step B. Only data for comparative t-tests were nonparametric, so a Mann-Whitney u test was run in place of a t-test.

Step C. T-tests were run to compare the means of geochemistry between seawater and EPF or between species. ANOVAs were run with pH as a four level factor. ANOVA and t-test significance was achieved if the p-value was less than 0.05.

Regression analysis was performed using the GraphPad Prism software and significance was denoted if the slope of the regression was statistically non-zero.

We revised the methods to clearly explain the statistical analyses.

Regarding experimental design we studied one calcitic species (*C. virginica*) that has ecological and economic importance, and is used in paleoclimate research. The aragonitic species whilst slow growing has been the subject of numerous geochemical and also metagenomic studies developing it as an archive of environmental information (eg. ; Schöne et al., 2005; Milano et al., 2017; Liu et al., 2015; Der Sarkissian et al., 2017). Importantly these specimens have a lot of supporting information generated and published in a prior study - for example the carbonate chemistry of the pallial fluid has been well-characterized (Downey-Wall, et al., 2022).

Specific comments:

Line 38 – insert "e.g.," in the beginning of brackets. Use this approach in other parts of the text where you are just presenting selected references for a certain statement.

Line 43 – replace "e.g.." with "e.g.,"

Response #3 We added e.g. to both bracketed reference lists.

Line 43 & 50 & 137 – not sure what you mean by "reviewed", why not just citing the reference

Response #4 We changed the text to cite the reference instead of writing "reviewed by".

Line 60 – place relevant references after mentioning individual species names and not at the end of the sentence.

Response #5 The relevant references were placed after each species name rather than all at the end.

Line 77 – did you mean Zhao et al. 2018? There is no publication from 2016 in reference list. Needs to be checked in detail.

Response #6 The correct reference was Zhao et al. 2017 which was added to the reference list.

Line 81 – a reference to the Figure should be before the Figure, move the Figure below this text

Response #7 We rearranged to put all figures after the reference.

Figure 1 – consider increasing a bit font site on the bottom of the right figure part

Line 88 – remove extra dot

Line 89 – check the presentation of isotope names and formatting of numbers

Lines 88-90 Species names should be presented in italic font

Response #8 All these minor revisions were corrected. All isotopic name formatting was fixed and species names were italicized.

Table 1 – the title should be presented above the table. Extra horizontal and vertical lines should be omitted from the Table for clarity. This table should not be presented in the Introduction section – it actually contains the Results of this study. It is not sufficiently clearly explained why parameters marked with n.d. were not measured.

Response #9 Figure was moved to the results section. We added that the data could not be measured due to sample size in the figure caption, which was explained in the methods section.

Line 96 – it is not clear why the authors refer here to Table 1 and Zhao et al. 2018 paper – and to which of them – as two are listed in the reference section. Did you want to refer to Figure 1? There you cite Zhao et al. 2016 paper.

Response #10 We changed it to just refer to Zhao et al 2018.

Line 109 – At the end there is only a reference to Crenshaw (1972), so check "et al."
Line 111 – year missing after Crenshaw
Line 143- 145 place references after corresponding taxa, not at the end of the sentence

Response #11 Minor revisions added.

Lines 150-160 & other parts of the text. There is no need to go in so much detail in relation to other taxa. You text is too wide and too descriptive. You need to focus on target taxa and main questions.

Response #12 In the shortened introduction will focused our text on our taxa or species of interest.

Line 164 Full species name is mentioned before in text, so abbreviations should be used here-
Response #13 We used the abbreviation after writing the full species name.

Line 176 Authors need to provide a short description of collection and culturing in this manuscript, it is not sufficient to refer to earlier manuscript.  There is no sufficient indication of year research was conducted.

Response #14 We included the dates of the collection and experiment dates.

Line 178 & other parts of the text – remove "psu"
Response #15 We removed psu from the text.

Line 182 you need to clearly specify size of bivalves used in your research as well as sample number

Response #16 This information was added to the revised methods and materials section.

Line 183 – check the formatting of geo coordinates
Response #17 Coordinates were reformatted.

Line 190 what do you ean by "see" here?
Response #18 "see" was removed from this reference.

Line 191 – here you refer to Table a that is presented several pages before – this is not appropriate approach to organisation of the manuscript
Response #19 We moved the table to the methods section.

Line 194 you need to define "dry weight". What did you measure & with what? What was the precission?

Line 195 You need to define "buoyant weight"

Response #20 We added a more specific explanation of buoyant weight and technique and calculations in the methods section.

Line 207 Arctica islandica is a slow growing species – you need to explain why you choose 14 days as period here.

Response #21 A better explanation is provided in the methods section. We were interested in the geochemistry of the shell and EPF of *A. islandica* specimens under ambient $CO_2$ conditions, which these organisms naturally grew as an aragonitic species geochemical point of comparison with the calcitic *C. virginica* data. As *A. islandica* was not subjected to an environmental challenge experiment we did not require the specimens to generate significant new growth under new experimental conditions for the purpose of the experiment, as was needed for the variable $CO_2$ experiment on *C. virginica*.

Line 212 You need to present Figure/diagram clearly illustrating how sampling of shell carbonate material was conducted.
Response #21 We replaced figure 1 with a figure to show how EPF sampling and geochemical sampling was conducted.

Line 232 A sentence should not start with a number, revise
Response #22 We edited this out.

Line 289 You refer here to Table 3 – and it is presented much later in the manuscript. There are serious issues with the organisation of your text, Figures and Tables.
Line 294 & other parts of text - Species names should be presented in italic font.
Line 304 – check the formatting of Fig – "a" vs. "A"
Response #23 location of the figure was fixed and formatting was also fixed.

Line 320 – statistical analysis used should be clearly presented in Material and Methods section. Did you test your data for homogeneity of variance (requirement of t test)? See for example Fig 3f – variances are not clearly not homogeneous according to your data presentation.

Response #24 Addressed in response #2.

Careful formatting of references is needed. Journal names are not consistently presented, as some are referred to by full names and others by their abbreviations. I did not check all in detail, and the following are just some observations: Lines 644-647 – check for use of capital letters in publication titles. Line 646 replace "mollusc" with "Mollusc". Line 652 – place "Porites" in italics, also check other references and present all species scientific names in italics font. Line 670 – year should be placed at the end of the reference for consistency. Line 677 – doi missing. Line 681 – capital letters are used, needs to be corrected.

Response #25 We carefully edited the references section which has a few organizational issues and lack of consistency.

References cited in response:
1. Schöne, B. R., Fiebig, J., Pfeiffer, M., Gleβ, R., Hickson, J., Johnson, A. L., ... & Oschmann, W. (2005). Climate records from a bivalved Methuselah (*Arctica islandica*, Mollusca; Iceland). Palaeogeography, Palaeoclimatology, Palaeoecology, 228(1-2), 130-148.
2. Milano, S., Nehrke, G., Wanamaker Jr, A. D., Ballesta-Artero, I., Brey, T., & Schöne, B. R. (2017). The effects of environment on *Arctica islandica* shell formation and architecture. Biogeosciences, 14(6), 1577-1591.
3. Liu, Y. W., Aciego, S. M., & Wanamaker Jr, A. D. (2015). Environmental controls on the boron and strontium isotopic composition of aragonite shell material of cultured *Arctica islandica*. Biogeosciences, 12(11), 3351-3368.
4. Der Sarkissian, C., Pichereau, V., Dupont, C., Ilsøe, P. C., Perrigault, M., Butler, P., ... & Orlando, L. (2017). Ancient DNA analysis identifies marine mollusc shells as new metagenomic archives of the past. Molecular Ecology Resources, 17(5), 835-853.
5. Downey-Wall, A. M., Cameron, L. P., Ford, B. M., McNally, E. M., Venkataraman, Y. R., Roberts, S. B., ... & Lotterhos, K. E. (2020). Ocean acidification induces subtle shifts in gene expression and DNA methylation in mantle tissue of the Eastern oyster (*Crassostrea virginica*). Frontiers in Marine Science, 7, 566419.
6. Zhao, L., Schöne, B. R., & Mertz-Kraus, R. (2017). Delineating the role of calcium in shell formation and elemental composition of *Corbicula fluminea* (Bivalvia). Hydrobiologia, 790, 259-272.

R2

Response #1 We thank reviewer 2 for their opinion that our work represents a major advance in biomineralogy research. We go on to address their specific comments below.

The temperatures for A. islandica are worryingly warm, near the upper range of this clam's thermal tolerance (lines 179/180; Seawater was maintained at a pH of 7.93 ± 0.09, temperature of 18.2 ± 1 o C, and salinity of 35 psu for the aragonitic clam A. islandica in the control conditions (Downey-Wall et al., 2020). The authors need to provide this as a caveat to the results. In other words, are the results scalable for all temperature ranges?? The

authors should consider the results of Liu et al. (2015) Environmental controls on the boron and strontium isotopic composition of aragonite shell material of cultured Arctica islandica, Biogeosciences, 12, 3351-3368, doi:10.5194/bg-12-3351-2015, whereby there seemed to be a potential influence of warmer temperatures on boron isotopes.

Response #2 Thank you for this point. *A islandica* and *C. virginica* specimens were maintained at different temperatures (9°C and 18°C, respectively) ; this was edited in the revised manuscript.

Add the length of time for the experimental calibration for both species in line 180 at the end.
Response #3 We added the length of time in the revision.

What are the ages and shell heights for the A. islandica shells? They grow very slowly, thus this is important to have these metrics in this study- (not just citing Downey-Wall et al. (2020))

"2.2 Calcification rate measurements Net calcification rate was calculated using the dry weight at the start and end of the experiment. Initial dry weight was measured at the start of exposure, on day 33 or 34, after the acclimation period (Downy-Wall et al., 2020). The buoyant weight was measured on either day 50 or 80 and the final dry weight was derived using a linear relationship between oyster dry weight and oyster buoyant weight (Ries et al., 2009)."

This may be suitable for juvenile mollusks but not for adults, especially A. islandica. What are the uncertainties in such measurements for large adult clams?

Why haven't the authors reported calcification rates for A. islandica. This is a central variable that needs to be considered (like Fig. 2a for oysters).

Response #4 We will clarify that calcification measurements were only conducted on *C. virginica* specimen, not *A. islandica*. Our calcification measurements were taken to understand how calcification was affected by ocean acidification treatments, which *A. islandica* were not exposed to.
Shell sizes were added to the revision of the materials and methods section and a better explanation was provided. We were interested in the geochemistry of the shell and EPF of *A. islandica* specimens under ambient $CO_2$ conditions. The *A. islandica* specimen grew under ambient $CO_2$ conditions as they were collected in their natural environment. Our need to maintain specimens under laboratory conditions was specifically to sample the EPF fluid. *A. islandica* is included in the paper to provide an aragonitic point of comparison in EPF chemistry to the calcitic *C. virginica* but as it only grew under ambient conditions and we were not examining its response to varying $CO_2$ we did not need to grow it sufficient duration to generate new growth.

The longer term experiment for *C. virginica* was needed to have the faster growing *C. virginica*, lay down new growth under different $CO_2$ treatments that can then be measured alongside the EPF chemistry. This experimental design is now explained in the methods section.

How are the authors confident that they sampled ONLY calcium carbonate reflecting the experiment? Did they stain the shells with calcein? Did they measure linear growth? This is most relevant for A. islandica because of relatively slow growth rates (i.e., see Liu et al., (2023) Resistant calcification responses of Arctica islandica clams under ocean acidification conditions, Journal of Experimental Marine Biology and Ecology, https://doi.org/10.1016/j.jembe.2022.151855.)

Response #5: To sample the new growth of *C. virginica* shells, we sampled a thin layer across the inner surface at the growing edge of the  shell, avoiding any parts such as repaired shell laid down to cover bore holes, as we thought the chemistry of those regions might be different. We don't have a marker such as a calcein stain to show where the new growth is, so we sampled shell material that was in close contact with the EPF and was recently laid down.

Shell sampling – the organic matrix in shells contain about a magnitude more boron than in the shell, and this likely has a very different isotopic composition (value). Are the authors confident all organics were removed?

Response #6 Yes, before isotopic analysis carbonate samples went through two oxidative cleaning steps to remove organics from the carbonate material. The oxidative agent consists of $H_2O_2$  which is similar to other studies (Pre-treatment effects on coral skeletal $\delta^{13}$C and $\delta^{18}$O - ScienceDirect). This cleaning was performed on various marine organisms and led to consistent data within species (corals, coralline algae, oyster, Liu et al. 2020). We can add a sentence before line 233 to explain this in more detail rather than just stating " 2.5-3.0 mg of oxidatively cleaned shell powders were dissolved in 1N HCl."

Why are the authors explain how they sampled the oyster shells but not the clam shells? The methods should have a parallel structure.

Response #7 The sampling was performed in the same way for both bivalve species, this was added into our methods section for clarity.

Very interesting result in Figure 3- showing different chemical composition of EPF compared to ambient seawater. Important finding that lots of folks have been suggesting but without the EPF evidence. And when you go to the shells even less Mg than seawater, and EPF. Thus the mollusks must be regulating calcifying fluids.

I really think the authors are missing an opportunity by not exploring changes in the shell geochemistry from both species here with growth rates, shell height, age, etc. The applicability/scalability of the study is far less without the inclusion of such metrics. Why not include these data?

Response #8 Some of those data were not collected for both species. We have calcification data for *C. virginica* however calcification was not recorded for *A. islandica*. Unfortunately, all specimens that were measured for calcification rate were used for a transcriptomic study (Downet-Wall et al., 2020) so comparisons between specimen's shell geochemistry and calcification were not possible

Ok- now some praise for the authors:

This is an important study with important implications. We learned that oysters (C. virginica) and clams (A. islandica) incorporate some elements and boron isotopes differently. The boron isotopic composition of the EPF for both species is different than seawater. The breakthrough of being able to sample the EPF chemistry/pH is a major advance in biomineralogy. Thus, a mechanistic model for biomineralization can be advance. Also, the mollusks evaluated here are not simple pH meters, and the shell d11B value is a mixture of the seawater d11B value and physiology. These results are consistent with in prep work that I am aware of now. Despite some issues with the description of the experiment and other concerns noted above, this is a major advancement.

References cited in response:
1. Grottoli, A. G., Rodrigues, L. J., Matthews, K. A., Palardy, J. E., & Gibb, O. T. (2005). Pre-treatment effects on coral skeletal δ13C and δ18O. Chemical Geology, 221(3-4), 225-242.
2. Liu, Y. W., Sutton, J. N., Ries, J. B., & Eagle, R. A. (2020). Regulation of calcification site pH is a polyphyletic but not always governing response to ocean acidification. Science advances, 6(5).
3. Downey-Wall, A. M., Cameron, L. P., Ford, B. M., McNally, E. M., Venkataraman, Y. R., Roberts, S. B., Ries, J. B., and Lotterhos, K. E.: Ocean acidification induces subtle shifts in gene expression and DNA methylation in mantle tissue of the Eastern oyster (Crassostrea virginica), Frontiers in Marine Science, 7, 566419, 2020.

---

## Author Response (AR2)

**Referee comments**

It is clear that Authors have put a lot of efforts in revising the manuscript, as its structure and content has been greatly improved. There are still several minor issues that Authors should address prior to acceptance. These are listed below:

Line 118 Provide shell lengths for Arctica islandica

Response 1: lengths of A. islandica added after line 118.

Line 128 Remove extra space between „31" and „."; Suggestion – provide full species name when it is at the beginning of the sentence (check also in other parts of the manuscript, e.g. lines 267, 329, etc.)

Response 2: The extra space was removed. When the species name started a sentence, the full scientific name was used instead *C. virginica* or *A. islandica*.

Line 138 Table title to go above table? Check also for other tables.
Line 139 Remove extra „."

Response 3: For both tables, the table title was placed above the table rather than below.

Line 160 vs 170, revise to avoid start of the both sentences with „Shells"

Response 4: Sentences were changed to not start with "shells."

346 should „(d)" be „(k)" – please check. „(j)" is also missing from legend, please check in detail Line 563 and 564

Response 5: Figure caption was revised to reference the correct graph.

Please revise a bit first and second sentence in conclusion, there is a certain overlap.

Response 6: The second sentence was removed because of the redundancy.

There are still issues with formatting of references, just several examples below:
616 Missing page numbers?

Response 7: Page number was added to the reference.

629 Put scientific name of species in italic

Response 1:Scientific names were italicized.

631 Remove capital letters where not needed

Response 1: The capitals were removed when not needed.

632 article number missing

Response 1: Article number was added.